# ACES: GENERATING DIVERSE PROGRAMMING PUZZLES WITH AUTOTELIC LANGUAGE MODELS AND SEMANTIC DESCRIPTORS

## ABSTRACT

Finding and selecting new and interesting problems to solve is at the heart of curiosity, science and innovation. We here study automated problem generation in the context of the open-ended space of python programming puzzles. Existing generative models often aim at modeling a reference distribution without any explicit diversity optimization. Other methods explicitly optimizing for diversity do so either in limited hand-coded representation spaces or in uninterpretable learned embedding spaces that may not align with human perceptions of interesting variations. With ACES (Autotelic Code Exploration via Semantic descriptors), we introduce a new autotelic generation method that leverages semantic descriptors produced by a large language model (LLM) to directly optimize for interesting diversity, as well as few-shot-based generation. Each puzzle is labeled along 10 dimensions, each capturing a programming skill required to solve it. ACES generates and pursues novel and feasible goals to explore that abstract semantic space, slowly discovering a diversity of solvable programming puzzles in any given run. Across a set of experiments, we show that ACES discovers a richer diversity of puzzles than existing diversity-maximizing algorithms as measured across a range of diversity metrics. We further study whether and in which conditions this diversity can translate into the successful training of puzzle solving models.

## 1 INTRODUCTION

Finding and selecting new and interesting problems to solve is at the heart of curiosity, science and innovation (Chu & Schulz, 2020; Schmidhuber, 2013; Herrmann et al., 2022). We propose to leverage machine learning, a set of tools usually targeted at *solving problems*, to automate the generation of an *interesting diversity of solvable problems*. Automated problem generation has a wide range of applications such as education (generating problems for students to solve), data augmentation (generating problems and solutions for AI model training), or automated scientific discoveries (e.g. discovering new scientific problems and their solutions).

In this work, we focus on the generation of a diversity of Python programming puzzles, an open-ended space to explore that contains problems ranging from trivial string manipulations to open mathematical puzzles (Schuster et al., 2021). Importantly, puzzle-solution pairs produced by the search can be checked for correctness using a Python interpreter, providing a notion of ground truth that natural language problems lack (e.g. creative writing). The automated generation of diverse programming puzzles could benefit computer science education and be used as a data generation process for the training of large language models (LLMs). Pretraining on code indeed seems to be a major factor in LLMs' reasoning abilities (Madaan et al., 2022; Liang et al., 2022; Fu et al., 2022).

Standard generative models do not explicitly optimize for diversity but are instead trained to fit the distribution of a reference dataset (e.g. Goodfellow et al., 2014; Brown et al., 2020; Chen et al., 2020; Ho et al., 2020). Measuring and optimizing for diversity requires the definition of a *behavioral characterization* (BC) of the generated artefacts on which to evaluate the measure. Early diversity-producing methods often used hand-coded low-dimensional representation functions, which focused and restricted the diversity search along features one could easily compute. More recent methods leverage pretrained embedding functions allowing them to work with higher-dimensional data

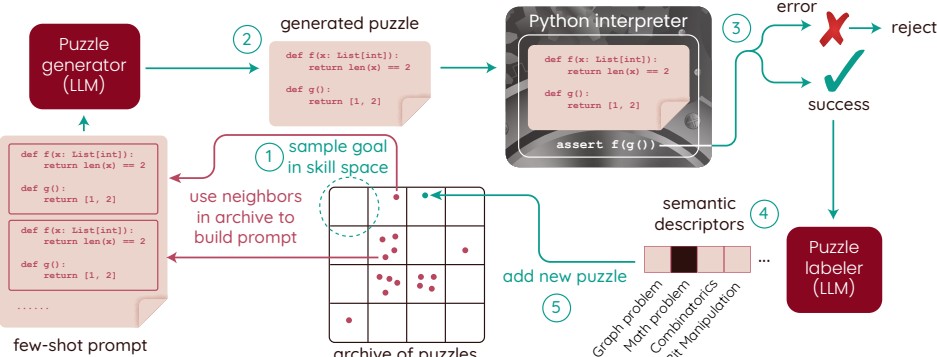

Figure 1: Overview of ACES. ACES maintains an archive of discovered puzzles grouped into cells indexed by their semantic representation (skill combination). ACES runs in several steps: 1) sample a target semantic goal and relevant examples from the archive. 2) given these, generate a puzzle f and its solution g with the puzzle generator. 3) test the validity of that pair by running assert(f(g()) in the interpreter. 4) if the pair is valid, obtain its semantic representation with the puzzle labeler. 5) add the new pair to its corresponding cell in the archive.

(e.g. image, text, programs) at the expense of interpretability and control over the axes of variations (Nair et al., 2018; Laversanne-Finot et al., 2018; Cully, 2019; Etcheverry et al., 2020).

We propose to leverage *semantic descriptors*: a hand-defined list of abstract features evaluated by LLMs. Semantic descriptors allow to work with high-dimensional inputs (here programs) while focusing the diversity search along interpretable semantic features of interest. Specifically, we represent any puzzle by the set of programming skills required to solve it among 10 possible skills (e.g. graphs, dynamic programming, recursion). Evaluating descriptors with LLMs allows us to define more abstract features that better capture our intuitive perception of axes of variation, descriptors that would have been hard or even impossible to code by hand. The compositionality of language and linguistic categories futher allow us to easily define sets of orthogonal conceptual categories that can be almost arbitrarily combined (Colas et al., 2020).

This work introduces a new diversity-producing algorithm called ACES for *Autotelic Code Exploration with Semantic descriptors*. ACES leverages an LLM for puzzle generation, solution generation and novelty evaluation. It slowly grows an *archive* of discovered puzzle-solution pairs, where each cell of the archive contains puzzles that share a given semantic representation — a 10D binary vector obtained by the semantic descriptors. At each new cycle of the algorithm, ACES targets a cell randomly in the archive (semantic goal) and generates a candidate puzzle and solution by prompting the LLM-based puzzle generator with the target semantic representation and a set of examples from the archive. The generated puzzle-solution pair is then evaluated for validity using a Python interpreter and, if valid, gets encoded by the puzzle labeler into a corresponding semantic representation used to store the newly discovered puzzle in the right archive cell, see Figure 1. Our experiments study the evolution of several diversity metrics over time and compares ACES with state-of-the-art baselines (Lehman et al., 2022; Haluptzok et al., 2023).

To summarize, our contributions in this paper are the following:

- We define the notion of *semantic descriptors* to leverage LLMs for the encoding of high-dimensional textual data into hard-to-compute, abstract and interpretable features of interest.

- We introduce a set of such semantic descriptors to characterize the diversity of programming puzzles based on classical programming ontologies.

- We propose *Autotelic Code Exploration with Semantic descriptors* (ACES), a new diversity-producing method building on these semantic descriptors that leverages the few-shot learning abilities of LLMs to generate an interesting diversity of programming puzzles;

- We evaluate the ability of ACES and its baselines to achieve various kinds of diversities and provide a comprehensive analysis of the interactions between diversity and finetuned performance on a held out test set.

## 2 RELATED WORK

Diversity-producing algorithms were originally proposed within the field of evolutionary computing. Beginning with *novelty search* (Lehman & Stanley, 2011b;a), this line of research expanded with the invention of quality-diversity algorithms (QD: Mouret & Clune, 2015a; Cully & Demiris, 2018a), a set of methods striving to evolve a diverse population of locally-performant individuals via the undirected mutation of existing solutions. A parallel line of research introduced *goal-directed* exploration processes, also called *autotelic learning*, where exploring agents learn to represent and sample their own goal as a way to direct the diversity search (Baranes & Oudeyer, 2013; Forestier et al., 2022; Colas et al., 2022). Although autotelic methods were first developed to model the open-ended development of children in skill learning robots Moulin-Frier et al. (2014); Oudeyer & Smith (2016), they have also proved effective in the automatic exploration of complex systems, either simulated (Reinke et al., 2019; Etcheverry et al., 2020) or physical (Grizou et al., 2020).

In all these methods, one must define a BC space to characterize novelty. The earliest works used predefined low-dimensional descriptors to represent generated artefacts (Lehman & Stanley, 2011b; Baranes & Oudeyer, 2013; Mouret & Clune, 2015b), which constrains the search along a handful of features one can code a descriptor for. More recent works have relied on higher-dimensional learned or pretrained embedding functions (Nair et al., 2018; Laversanne-Finot et al., 2018; Reinke et al., 2020), and even hierarchies of such spaces, each representing different perceptual features of the generated artefacts (Cully & Demiris, 2018b; Etcheverry et al., 2020). Diversity-search algorithms sometimes need to be adapted to work with such high-dimensional spaces whose discretization leads to an exponential number of cells (Vassiliades et al., 2017). But the main issue is that they are hardly interpretable and might not always align with the dimensions of variation humans find meaningful. With ACES, we propose an autotelic diversity-producing algorithm that constrains the search along a set of abstract, interpretable and hard-to-compute features of interest evaluated by LLMs.

This work follows of a recent trend on leveraging feedback or generations from LLMs to improve older learning architectures. The original inspirations for this paper come from the QD literature, where LLMs are now used to suggest mutations and crossover in diversity-producing evolutionary algorithms (Lehman et al., 2022; Bradley et al., 2023b; Meyerson et al., 2023). Several approaches also rely on LLMs to replace human feedback (AI feedback): e.g. to finetune other LLM models (Bai et al., 2022; Lee et al., 2023), to characterize generated poems (Bradley et al., 2023a), to revise the policy of autotelic agents (Wang et al., 2023a), to suggest goals for them (Colas et al., 2023; Du et al., 2023) or measure their *interestingness* (Zhang et al., 2023). With ACES, we use AI feedback to compute abstract and intepretable representations of programming puzzles so as to optimize for diversity in that space. In a similar way. QDAIF (parallel work) uses 2D LLM-generated characterisations in the context of poem generation, a space where there is not such a clear notion of feasibility and solvability (Bradley et al., 2023a).

## 3 METHODS

We first present the *programming puzzles* (Section 3.1) and discuss measures of *interesting diversity* (Section 3.2). Then, we introduce ACES, our new method for diversity generation (Section 3.3) and define relevant baselines (Section 3.4). We will open-source the implementation of the algorithms and the datasets of generated puzzles and solutions with the camera-ready version.

### 3.1 PROGRAMMING PUZZLES AND THE P3 DATASET.

The *Python Programming Puzzles dataset* (P3) contains 1715 puzartefactszle-solution pairs where each puzzle is defined by a short test program `f` designed to verify the validity of solution programs `g` such that valid solutions satisfy `f(g()) == True` when run in an interpreter, see example in Figure 2 (Schuster et al., 2021). P3 puzzles span problems of various difficulties that involve different programming skills: e.g. string manipulation, classic (e.g. Tower of Hanoi) and more complex programming problems (e.g. involving dynamic programming or factoring), or even open problems in computer science or mathematics. The P3 dataset is split into training and testing datasets ($N = 636$ and 1079 respectively). Traditionally, a solver model is trained on puzzle-solution pairs from the train set and evaluated on the test set. Both datasets are pre-filtered to examples shorter than 1024 tokens to accommodate for restricted context windows.

```python
def f(ls: List[str]):
    """Divide the decimal representation of 8^88 up into strings of
        length eight."""
    return "".join(ls)==str(8**88) and all(len(s)==8 for s in ls)
def g():
    return [str(8**88)[i:i+8] for i in range(0,80,8)]
assert f(g()) == True
```

Figure 2: Example of a simple programming puzzle and its solution from the P3 dataset (Schuster et al., 2021). A solution function `g` must return a valid solution such that `f(g()) == True`.

## 3.2 MEASURING INTERESTINGNESS AND DIVERSITY

Diversity search aims to generate collections of artefacts that are both *diverse* and *interesting*, two *subjective* measures that strongly depend on the observer's point of view. Let's define them.

**Defining interesting representation spaces.** One can generate interesting diversity by generating it in a representation space where *everything is interesting*, meaning that all uninteresting samples collapse to small regions. This requires the careful definition of a *representation function $R$* mapping each artefact $p$ (here puzzle) to a numerical representation $z_p = R(p)$ and a *metric $m$* to compute distances between them. What should these functions be in the context of our programming puzzles?

First, we use cosine distance computed in three different continuous embedding spaces — a standard approach for representing programs and text in general (Reimers & Gurevych, 2019). Here, we use *salesforce/codet5p-110m-embedding* (Wang et al., 2023c), *wizardlm/wizardcoder-1b-v1.0* and *wizardlm/wizardcoder-3b-v1.0* (Luo et al., 2023) embedding models from the HuggingFace's Hub (Wolf et al., 2020) to obtain 256D, 2048D, and 2816D continuous embedding representation vectors.

Second, we propose to represent programming puzzles using *semantic descriptors* — a set of hand-defined features selected from a standard computer science textbook to capture interesting differences in the programming puzzles (Cormen, 2009). We define 10 of these: Sorting and Searching, Counting and Combinatorics, Tree and Graph problems, Mathematical Foundations, Bit Manipulation, String Manipulation, Geometry and Grid Problems, Recursion and Dynamic Programming, Stacks and Queues, Optimization Algorithms, see definitions in Appendix Section A.2. A puzzle is then represented as a 10D binary semantic vector $z_p$, where each value $z_p^i$ evaluates whether the puzzle requires skill $s_i$ (1) or not (0) to be solved. It is unclear how we could write a piece of code to label puzzles along these dimensions. Instead, we ask ChatGPT to assign these labels (version *gpt-3.5-turbo-0613*). In the example of Figure 2, ChatGPT assigns labels for *Sorting and Searching*, *Counting and Combinatorics* as well as *String Manipulation* (encoding 1100010000, see other examples in Appendix Section A.2). We use Hamming distance in that semantic space.

The produced diversity will necessarily be shaped and constrained by the subjective choice of the representation function $R$. A good representation function conflates uninteresting objects in small areas of the representation space (e.g. all puzzles that require no skills are mapped to the 0000000000 cell of our semantic space) and "spreads out" interesting objects. In such a space, most uninteresting puzzles look the same (low inter-puzzle distance) while interesting puzzles look different (high inter-puzzle distance) and optimizing for diversity leads to the generation of diverse interesting puzzles. The experimenter controls specifies semantic descriptors of interests and thus controls the resulting diversity produced. Optimizing for diversity in pretrained embedding spaces is also a subjective choice (different embedding representation will lead to different diversities), but it is made implicitly: the experimenter does not really know what they are signing for.

This semantic representation function is a contribution of this paper. We hypothesize that it is more aligned with human perception of programming puzzles than continuous embedding functions. Our proposed algorithms are designed to maximize this form of diversity (see Section 3.3). We hypothesize that this will achieve not only higher diversity scores in this semantic representation space, but also higher scores in the continuous embedding representation spaces.

**Measuring diversity.**    We measure the diversity of sets of generated puzzles in different ways. We use counts of discovered puzzles and cells: 1) the number of discovered cells (filled with at least 1 puzzle), 2) the number of cells discovered beyond the ones covered by the train set, 3) the number of valid puzzles that were generated, 4) the number of valid puzzles generated beyond the cells covered by the train set. We also track measures of density or entropy: 5) the average pairwise distance between embedding representations, 6) the entropy of the distribution of semantic representations.

**An utilitarian take on measuring interesting diversity.**    Interesting puzzles must be solvable, which is why we filter out invalid puzzle-solution pairs. One could also be interested in training a puzzle solver to achieve high-performance on a specific problem distribution. In this case, we would perceive a collection of generated puzzle-solution pairs as *more interesting than others* if the solver finetuned on this set outperforms the same solver finetuned on the other sets when tested on the target distribution (e.g. P3's test set). Section 4.4 will look at correlations between various metrics and the final performance of a LLaMA model (*openlm-research/open_llama_3b_v2* on HF's hub, Geng & Liu, 2023) after finetuning for two epochs on the generated set of puzzle-solutions. In line with previous research (Chen et al., 2021), we will report the Pass@k performance metric on the testing set of P3 for $k \in [1..10]$: the percentages of puzzles for which at least one valid solution is generated within $k$ attempts. In addition to the diversity metrics listed above, we will look at various metrics measuring how well the generated set of puzzle-solution pairs covers the testing distribution.

## 3.3    AUTOTELIC GENERATION OF INTERESTING DIVERSE PUZZLES

This section introduces ACES, a new diversity-producing algorithm that generates an interesting diversity of programming puzzles by optimizing for the novelty of each new generation in the semantic space described above, see Figure 1. ACES grows an *archive* of diverse puzzle-solution pairs. It repeatedly: samples a semantic goal from the archive, generates a new puzzle-solution pair conditioned on that goal and, if the pair is valid, labels and adds it to the archive. We use the *Chat-GPT* LLM (*gpt-3.5-turbo-0613,* Schulman et al.). Algorithm 1, Figure 1 and Appendix Section A.2 respectively present the pseudo-code, illustration and prompts of ACES.

---

**Algorithm 1:** Pseudo-code of ACES

---

Initialize an archive $\mathcal{A}$ `(with labeled puzzle-solution pairs from the P3 train set)`
**for** $i = 1$ **to** $N$ **do**
> Sample a goal: $z_g \sim \text{Uniform}(\mathcal{A})$ `(uniform sample of a semantic goal)`
> Sample examples: $e \sim E(\mathcal{A}, z_g)$ `(nearest neighbor sampling with Hamming distance)`
> Generate puzzle and solution: $(p, s) \sim \text{LLM}(\text{prompt}_{\text{gen}}(g, e))$ `(see Appendix A.2)`
> Test puzzle-solution pair: pass $= \text{p}(\text{s}())\text{==True}$ `(using the interpreter)`
> **if** *pass* **then**
>> Label the puzzle $z_p \sim \text{LLM}(\text{prompt}_{\text{lab}}, p)$ `(see Appendix A.2)`
>> Add $(p, s, z_p)$ to the archive $\mathcal{A}$ in cell $c_{z_p}$

---

**Sampling a goal and relevant examples.**    ACES selects a semantic goal by sampling uniformly among the set of $2^{10}$ possible semantic representations. We then greedily select three closest examples from the archive using the Hamming distance in the semantic space: two from the generated puzzle-solution pairs and one from P3's train set to always keep a well-formatted puzzle example.

**Puzzle generator.**    The puzzle generator is implemented by an LLM. Conditioned on the semantic goal and the three examples, we ask the LLM to generate a puzzle-solution pair that would be labeled with the target semantic vector. For each *cycle* of the algorithm and to leverage the parallelization of LLM calls, we repeat the process of sampling a goal, examples, puzzles and solutions 10 times before considering the addition of these candidates to the archive. Using a Python interpreter, we filter the set of valid puzzle-solution pairs and send them to the puzzle labeler.

**Puzzle labeler.**    The puzzle labeler computes the semantic representation vector of each valid puzzle-solution pair. The prompt details the task to the LLM and presents the complete list of skills, then asks it to compute its semantic representation (see Section 3.2). The puzzle-solution pair is finally added to its corresponding cell in the archive. Note that, although we aim for a particular target semantic representation, whether we achieve the goal or not is not that important. What is

important is that the generated puzzle is valid and falls into a new cell. This is the driving principle behind the *hindsight learning* (Andrychowicz et al., 2017).

**What's new?** In addition of the semantic descriptors (whose novelty is discussed in Section 3.2), ACES is the first algorithm to leverage a goal-directed LLM for the generation of diverse artefacts via in-context learning. The LLM is here used as the generation engine and is steered towards generating novel and interesting puzzles via its goal-directedness and example selection (in-context learning). The algorithm *ELM* already used an LLM to suggest mutations of existing artefacts (Lehman et al., 2022). But like other quality-diversity algorithms, it is not goal-directed: it samples a previously-generated artefact from its archive, mutates it with the LLM in the hope of generating a new artefact that would fill a new cell.

### 3.4 BASELINES

**Static generative model (Static Gen).** This baseline was proposed in Haluptzok et al. (2023): it repeatedly prompts the LLM to generate a new puzzle-solution pair given three examples uniformly sampled from P3's train set.

**Ablation of goal-directedness (ELM semantic).** Instead of sampling a goal and asking the puzzle generator to reach it, we uniformly sample two puzzle-solution pairs from the archive that serve as examples. We then sample a cell in the archive and a puzzle from that cell, and ask the language model to output a mutation of this puzzle. The resulting algorithm is not autotelic anymore but becomes a variant of the QD algorithm *Map-Elites* (Mouret & Clune, 2015b). In fact, this implementation is a variant of the *ELM* algorithm (Lehman et al., 2022) where the explored representation space is our proposed semantic space.

**Ablation of goal-directedness and semantic representations (ELM).** We can further ablate ACES by removing the use of the semantic representation space. Instead, this baseline uses the continuous embedding space described in Section 3.2 (*Salesforce/codet5p-110m-embedding*, Wang et al., 2023c). This ablation is a variant of ELM (Lehman et al., 2022) where the explored representation space is a pretrained embedding space. To define a limited number of cells in this high-dimensional space, we use the method proposed in *CVT-Map-Elites*, a variant of MapElites that uses centroidal Voronoi tessallations (CVTs) to partition the space into a tractable number of well-distributed cells (Vassiliades et al., 2017). The partition is conducted in two steps. We first sample with replacement 40k puzzles from P3's train set and perturb their embeddings with a Gaussian noise ($\mathcal{N}(\mu = 0,\ \sigma^2 = 1.2)$) before normalizing them to unit-length. Then, we use the K-means algorithm (Steinhaus, 1957; MacQueen, 1967) to identify 1024 centroids and obtain the same number of cells as ACES in the archive. Once this archive is created, we simply run the ELM algorithm. ELM and ELM-semantic share their mutation operator but differ in the way the archive is maintained (CVT archive on continuous embedding features for ELM, archive of semantic cells for ELM-semantic).

**Generating an interesting diversity of textual artefacts.** Although the current paper focuses puzzle generation, ACES can in principle be used to generate an interesting diversity of any type of textual artefacts. For each new type of textual artefacts we want to generate a diversity of, we need to provide a set of features of interest the LLM will be able to evaluate on each new generation. Natural language provides a rich and flexible descriptive space. Compared to traditional representation functions that rely on hand-engineered features, the abstract nature of language allows us to describe the semantic qualities of textual artefacts in an open-ended way.

## 4 RESULTS

### 4.1 IS LLM LABELING FAITHFUL?

Our proposition of leveraging LLMs to semantically characterize generated programming puzzles is only meaningful to the extent that the LLM faithfully labels the puzzles. To make sure this is the case on the distribution of puzzles we end up generating, we compare the LLM-generated labels to hand-defined labels on a set of 80 puzzles sampled from the generated set of the seed of ACES

which has the highest label diversity. Details of how the puzzles were sampled for the computation of the confusion matrix can be found in Appendix Section A.3.

Figure 3 reports the confusion matrix and the number of puzzles containing the ground truth label for each row. The puzzle labeler demonstrates high true negative rates on most dimensions but sometimes struggles to detect present skills (low true positive rate), e.g. for the Stacks and Queues and the Geometry and Grid dimensions. Note that annotating puzzle with semantic labels is also hard for humans and that the classification does not need to be perfect to drive diversity (see Section 4.2). However, this poses a challenge when using the same labeler for evaluation purposes. We thus choose to report diversity metrics in three different embedding spaces that were not used for training.

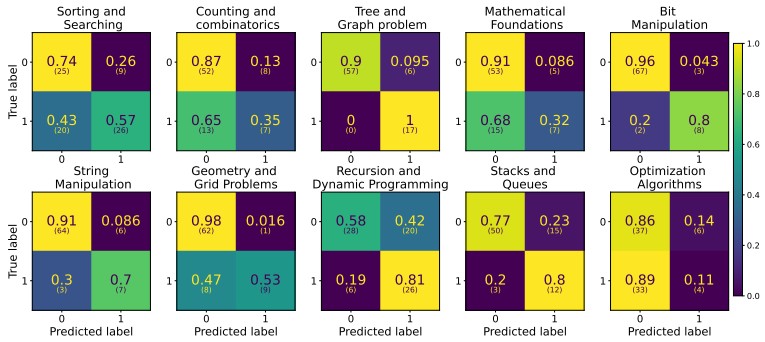

Figure 3: Faithfulness of semantic labeling. Confusion matrices for the multi-label classification task performed by the puzzle labeler. For each semantic descriptor, we report the confusion matrix where rows indicate the ground truth presence (1) or absence (0) of the skill while the column indicates its detection (1) or non-detection (0). We thus read from top left to bottom right: true negative, false positive, false negative, true positive rates (sample size in parenthesis).

## 4.2 MEASURING DIVERSITY

**Diversity in semantic space.** Figures 4a to 4e report the evolution of various diversity measures as a function of the number of puzzle-solution pairs generated by the puzzle generator. Semantic algorithms (ACES and ELM semantic) better explore the semantic representation space: they discover more cells beyond the cells covered by P3's train set (4b), more cells in general (4a) and generate more puzzles beyond the cells covered by the train set (4d) and their cell distributions have higher entropy (4e). ELM algorithms generate more valid puzzles in general, but the non-semantic ELM mostly generates puzzles falling in cells covered by the train set (4c vs 4d). Our goal-directed ACES generates puzzles whose cell distribution has higher entropy than other baselines (4e). Algorithms that optimize for diversity in the semantic space were expected to achieve higher diversity in that space, but does it translate to higher diversity in other representation spaces?

**Diversity in embedding spaces.** Figure 5a to 5c report a diversity metric (Friedman & Dieng, 2022) computed over three different embedding spaces (see Section 3.2). ELM demonstrates relatively high diversity in the embedding space it uses for optimization (5a) but lower diversity on other embedding spaces (5b, 5c). ACES achieves the highest diversity in the two WizardCoder embedding spaces (5b, 5c) while ELM semantic reaches the highest diversity in the Code5P embedding space (5a). These results demonstrate that optimizing for diversity in our custom semantic space also leads to higher diversity in other representation spaces.

## 4.3 QUALITATIVE INSPECTION OF SAMPLES

We here describe the most remarkable trends that we have observed by manually inspecting the data (see Appendix A.4). One tendency of the generation process across all experiments is to shift the definition of what a puzzle is. In the original formulation, the problem `f` implements a test that verifies a certain number of conditions are met, and `g` implements an *algorithm* that produces a value that satisfies the conditions. What the generation processes do in many cases is shift the algorithmic load from `g` to `f`, in which case `g` only provides arguments for `f`. We hypothesise this originally

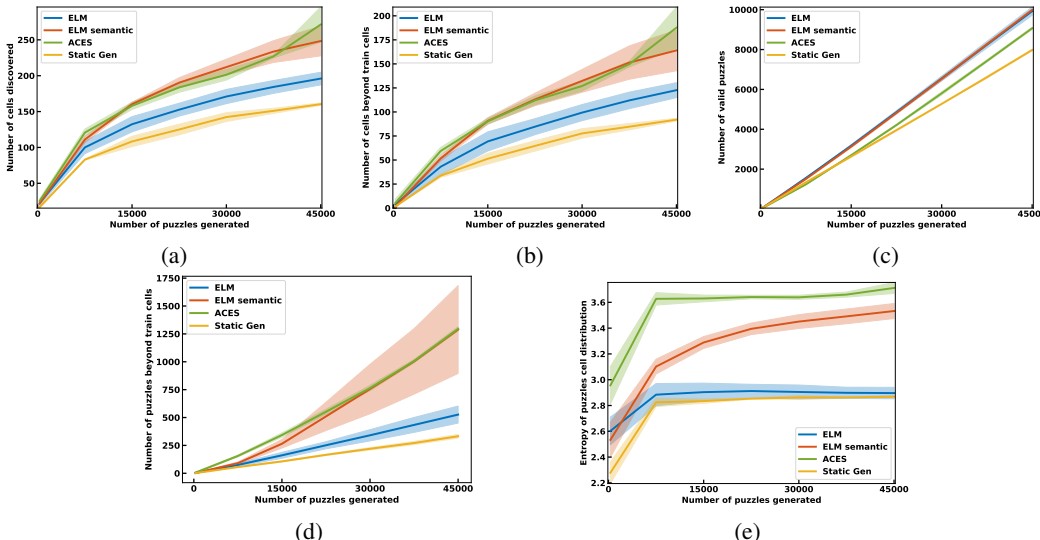

(a)  (b)  (c)

(d)  (e)

Figure 4: Diversity of generated puzzles in semantic space. We report the evolution of several diversity metrics computed in the semantic space as a function of the number of puzzle-solution pairs generated by the puzzle generator. Semantic algorithms (ACES and ELM semantic) achieve higher diversity in the semantic space.

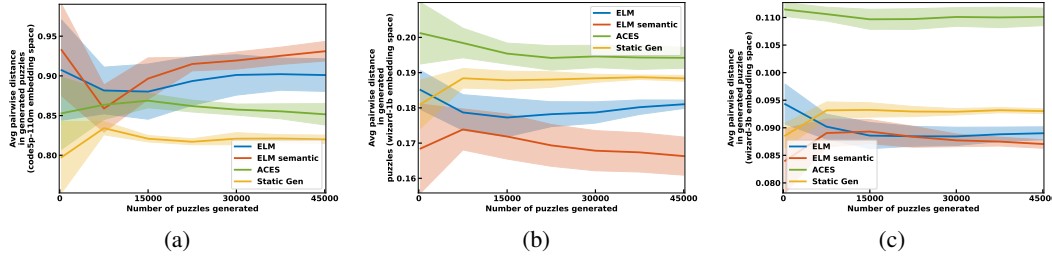

(a)  (b)  (c)

Figure 5: Diversity of generated puzzles in embedding spaces. We report the evolution of the pairwise distance between puzzle-solution pair embeddings as a function of the number of generated puzzle-solution pairs, for three different embedding representation spaces (average across seeds).

comes from the docstring description of puzzles in the seed examples, which are included in `f` but describe the behavior of `g`. Another important difference is what the puzzles look like: puzzles generated by the Static Gen baseline tend to be short and more math-heavy, while samples generated by ACES tend to be longer and more creative (see Appendix A.4 again for examples). Appendix Section A.5 provides additional visualizations: 2D projections of the generated puzzles embeddings using UMAP (Appendix Figures 11 to 14), depictions of the evolution of the archives as a function of time (Appendix Figure 10), as well as histograms for the distribution of skills and number of skills across generated puzzles for each of the algorithms (Appendix Figures 7). The UMAP projections, in particular, give a good sense of the difference in distribution between methods.

## 4.4 LOOKING FOR PERFORMANCE PREDICTORS

Our contributions lead to larger diversities of interesting programming puzzles. Could this translate into higher performance on an arbitrary target distribution we might be interested in? We consider P3's testing set of puzzles to be our target distribution and measure the Pass@k performance of LLaMA 3B models finetuned on the generated sets of puzzle-solution pairs.

Figure 6 shows Pass@k metrics for various values of $k$. *Static Gen* performs higher despite having the lowest diversity of all algorithms on all considered metrics. Other algorithms spend more time exploring the full-extent of the space of programming puzzles (see Figures 4 and 5) and thus less time focusing on generating puzzles close to the training distribution. *Generating more puzzle-solution pairs and a higher diversity of them does not lead to higher performance in our setup*. Note

that this can be perfectly fine. Our algorithms did not optimize for final performance on an arbitrary target distribution (here P3's test set) but optimize for pure diversity.

This said, we might be interested in figuring out how to constrain diversity-search to maximize performance on a target test distribution down the line. We thus test for correlations between the Pass@10 performance and both diversity metrics and metrics assessing the distance between generated puzzle-solution pairs from P3's test set (target distribution coverage). Over the test metrics we only found: an anti-correlation with the number of valid puzzles generated, an anti-correlation with the number of cells discovered and an anti-correlation with the average pairwise distance between generated puzzles in the Code5P embedding space, all mostly driven by *Static Gen*'s low numbers and high performance. The FID score in embedding space (Heusel et al., 2017), number of puzzles in cells

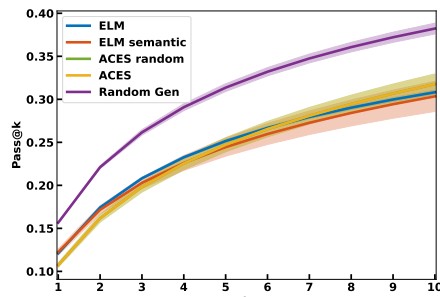

Figure 6: Downstream performance on P3's test set. Pass@k is the fraction of puzzles solved after $k$ attempts ($k \in$ [1:10]). Green overlaps with yellow.

covered by the test set, number of test cells discovered or the average distance between test puzzles and their nearest neighbors in the generated sets computing in embedding space all measure how much the generated puzzle-solution pairs cover the test distribution of puzzles but none of them were found to be significantly correlated with the downstream performance metric.

## 5    DISCUSSION

This paper introduced ACES, an autotelic generative algorithm that optimizes for diversity in a custom semantic description space adapted for the generation of programming puzzles. ACES produces stronger diversity in semantic and several embedding spaces, which results in the generation of interesting and creative puzzles as detected by manual inspection (Section 4.3).

Our last set of experiments uncovered a counter-intuitive result calling for more research: generating more puzzles of higher diversity does not translate into the higher downstream performance of an LLM finetuned on the generated data as measured over a target distribution of problems (here P3's test set). Surprisingly, we could not find any significant correlation between downstream performance and metrics measuring how much the generated set covers the target distribution. These results might be explained by more superficial drifts between the generated data and the test data: e.g. by the shift of the computational burden from the solution function to the problem function exposed in Section 4.3. This raises questions for future research: what are causal predictors of final downstream performance? Can we design the goal sampler of our autotelic diversity search to both search for maximal diversity while exploring the space of puzzle that will lead to good downstream performance? Can we characterize the trade-off between diversity and downstream performance?

Future work could also improve various aspects of ACES. One could replace the default uniform goal sampling with more sophisticated autotelic sampling methods (e.g. using novelty or learning progress intrinsic motivations, Colas et al., 2022) or improve on the selection of in-context examples to help the puzzle generator. ACES currently explores a fixed set of semantic features and is thus somewhat constrained to *combinatorial* forms of creativity (Boden, 1998). Moving forward, we could give it the ability to come up with new semantic features or prune others as the exploratory process unfolds, opening the door to more *exploratory* and *transformational* forms of creativity or even *historical* creativity if this system were to interact with human cultural evolution processes, as defined in Boden's work on human and artificial creativity (Boden, 1996; 1998).

Pure diversity-search is useful beyond its data augmentation applications. On the first end, it could be used to generate diverse and interesting problems to educate the new generations of programmers. It could also be combined with autotelic solving systems where it would optimize for both interesting diversity and *quality*. High-quality problems could for instance be the ones that are *intermediately difficult* for the learner (Florensa et al., 2018), or those that maximize its learning progress (Oudeyer & Kaplan, 2007). This paves the way for collaborative processes that endlessly co-evolve new interesting problems and their solutions, open-ended discovery systems that could be helpful for science (e.g. automatic theorem discovery and proving).

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

## REPRODUCIBILITY

Our experiments rely on ChatGPT (version *gpt-3.5-turbo-0613*) and will therefore be reproducible to the extent that this model stays available through the OpenAI API (OpenAI, 2023). This said, we expect our results to gain in impact and significance as the base LLM performance improves, which seems to be the current trend (e.g. GPT-4). Indeed, as the puzzle labeler improves, the labeling will become more and more reliable and closer to human-generated labels. As the puzzle generator improves, the semantic goals will be achieved more reliably, which will drive a better exploration. The implementations of ACES and other baselines, as well as the script to generate figures will be made available on github with the camera-ready version of this paper.

ETHICS

The experiments presented in this work did not involve any human participants and the authors declare no conflicts of interest. Methods building on ours can be applied in educational settings, in which case the generated samples need to be validated as being appropriate.

A more general comment on autotelic and open-ended methods is that they could lead to the creation of potentially harmful artefacts through the open-ended exploration process. This can be mitigated by methods to control the distribution of generated artefacts, but this increases the chance that ill-intentioned actors could use the technology.

## A APPENDIX

### A.1 ADDITIONAL RELATED WORK

A recent line of work has begun using large models as evolution operators in Neural Architecture Search (Chen et al., 2023; Nasir et al., 2023).

This work is also linked to prior approaches to generate synthetic data for models in language and code modalities. The seminal work here is by Honovich et al. (2023), whose technique is not autotelic and is stationary, not reusing prior generated data (related work in the code domain is Haluptzok et al. (2022); Roziere et al. (2023)). A step in the direction of non-stationarity is Self-Instruct (Wang et al., 2023b), which does not do any goal targeting. There are goal-conditioned approaches to synthetic data generation (Eldan & Li, 2023) which do not involve any data evolution and mainly build on concrete goals whereas we use abstract ones. The generation process for the Phi-1 model (Gunasekar et al., 2023) seems related but few details are given in the paper. Finally, Evol-Instruct Xu et al. (2023); Luo et al. (2023) is an evolutionary method that uses several prompts (with more abstract instruction like increasing difficulty) to evolve a set of instructions. It does not perform goal targeting in an abstract space, but its use of abstract instruction is related to targeting goals in our semantic space. None of these approaches perform explicit diversity maximization, instead, they rely on their inherent generation mechanisms; and many do not perform extensive diversity evaluation.

### A.2 PROMPTS

Here are the various prompts we use for ACES and all baselines.

**Skills description.** Skills description used to label problem. see prompt A.2

---

**Skills description**

0 - **Sorting and Searching**: Sorting refers to arranging a collection of elements in a specific order, typically in ascending or descending order. Searching involves finding the location or presence of a particular element in a collection.
1 - **Counting and Combinatorics**: Understanding principles of counting and combinatorial analysis, including permutations, combinations, and other counting techniques. These skills are essential for solving problems that involve counting the number of possibilities or arrangements.
2 - **Trees and Graphs**: Analyzing and solving problems related to tree and graph structures involving nodes connected by edges. This includes tasks such as traversal, finding shortest paths, detecting cycles, and determining connectivity between nodes.
3 - **Mathematical Foundations**: Strong understanding of mathematical concepts such as summations, probability, arithmetics, and matrices.
4 - **Bit Manipulation**: Performing operations at the bit level to solve problems.
5 - **String Manipulation**: Operations and algorithms specifically designed for working with strings. This includes tasks like concatenation, searching, replacing, and parsing strings.
6 - **Geometry and Grid Problems**: Understanding geometric concepts and algorithms for problem-solving, including grid-related problems. This involves tasks such as grid traversal, finding distances, detecting patterns, and solving geometric problems on grids.
mybox 7 - **Recursion and Dynamic Programming**: Utilizing recursive techniques and

---

dynamic programming approaches to solve problems by breaking them down into smaller subproblems and building solutions incrementally.

8 - **Stacks and Queues**: Data structures used to store and retrieve elements in a specific order. Stacks follow Last-In-First-Out, while queues follow First-In-First-Out. They are used for managing function calls, recursion, and implementing search algorithms.

9 - **Optimization Algorithms**: These algorithms aim to find the best possible solution for a given problem by minimizing or maximizing an objective function. They involve searching for optimal values within a given solution space, considering various constraints and parameters. For example, brute-force search (checks all possible solutions to a problem without using heuristics) and greedy search (locally optimal choices at each step to find the best solution) are examples of optimization algorithms in this category.

**Prompt for the puzzle labeler.** This prompt is used for ACES and all variants involving semantic representations.

---

### Puzzle labelling

**User:** I will give you a Python programming puzzle f (and its solution g) and a list of programming skills. Your role is to say which programming skills are required to understand and solve the problem.
skills:

**[Skills description]**

The puzzle is:

**[Insert Puzzle to label here]**

After completing your reasoning (you can start by explaining the problem and the solution in a few sentences). Ensure you remove every listed skills that are unnecessary for understanding or solving the given problem. It is necessary to summarize your answer by writing every index of categories explicitly used in the problem or solution in a Python list, following the format provided below. Please ensure the correct usage of the following text where <Python list> is a list with numbers from 0 to 9: "Therefore, the list of indices for the problem is: <Python list>"

**ChatGPT:**

---

### Example of puzzle labelling

**Puzzle to label:**

```
def f(s: str) -> bool:\\
    """Find a string with 1000 'o's but no two adjacent 'o's.
        """\\
    return s.count('o') == 1000 and s.count('oo') == 0\\
def g():\\
    return ('h' + 'o') * 1000\\
assert f(g())\\
```

**ChatGPT:** To solve the problem, we need to understand the logic of the 'f' function and how it checks for the presence of 1000 'o's and no adjacent 'o's. We also need to understand the implementation of the 'g' function, which generates the string with 1000 'o's.

Based on this understanding, the necessary programming skills are:
- 1: Counting and combinatorics (to understand the counting of 'o's and 'oo's in the string)

- 5: String Manipulation (to understand the string concatenation and searching)

Therefore, the list of indices for the problem is: [1, 5]

**Prompt for the puzzle generator of ACES.** This prompt is used for ACES and all autotelic variants.

---

**ACES**

**User:**
I will give you 3 (Puzzle 0 to Puzzle 2) Python Programming Puzzle (P3). A P3 consists of a problem f and its corresponding solution g. The puzzle is solved if f(g()) == True. Your role is to generate new puzzles according to the instructions given.
In addition each of those puzzles are associated with a list of skills. Here is a detailed description of those skills:

**[Skills description]**

Your role is to generate 3 new puzzles (Puzzle 3 to Puzzle 5) that require those skills: **[Insert list skills to target]**.
Note that the first argument of f is the output g(). Make sure to define and set values for all arguments of the function 'f' (excluding the first argument, as it is the solution that needs to be found and given by g). Both functions, 'f' and 'g' should have matching argument signatures: def f(arg0, arg1=value1, arg2=value2, ...) and def g(arg1=value1, arg2=value2, ...). Please provide all values (value1, value2, ... ) for all arguments. For example f(solution,arg1=1, arg2=2, ...) and g(arg1=1, arg2=2, ...). And you should not use f inside g.
Additionally, make sure to import any necessary libraries to ensure your code runs smoothly. Please ensure the mutated puzzles fall into all those skills: **[Insert list skills to target]**.
—-
Puzzle 0, required skills **[Insert list of skills associated with Puzzle 0]**:
**[Insert Puzzle 0]**
—
Puzzle 1, required skills **[Insert list of skills associated with Puzzle 1]** :
**[Insert Puzzle 1]**
—
Puzzle 2, required skills **[Insert list of skills associated with Puzzle 2]**:
**[Insert Puzzle 2]**
—

Could you please write 3 new interesting correct Python Programming Puzzles (from Puzzle 3 to Puzzle 5)? Please, ensure the new puzzles must necessitate the utilization of the following skills (required skills **[Insert list skills to target]**):
**[index skill 1 to target - Name of the skill 1 targeted**
**[index skill 2 to target - Name of the skill 2 targeted]**
.
.
.

**ChatGPT:**

---

**Prompt for the puzzle generator of Static gen.**

**Static gen**

**User:** I will give you 3 (Puzzle 0 to Puzzle 2) Python Programming Puzzle (P3). A P3 consists of a problem f and its corresponding solution g. The puzzle is solved if f(g()) == True. Your role is to write 3 new puzzles (Puzzle 3 to Puzzle 5). Note that the first argument of f is the output g(). Make sure to define and set values for all arguments of the function 'f' (excluding the first argument, as it is the solution that needs to be found and given by g). Both functions, 'f' and 'g' should have matching argument signatures: def f(arg0,

arg1=value1, arg2=value2, ...) and def g(arg1=value1, arg2=value2, ...). Please provide all values (value1, value2, ... ) for all arguments. For example f(solution,arg1=1, arg2=2, ...) and g(arg1=1, arg2=2, ...). And you should not use f inside g.
Additionally, make sure to import any necessary libraries to ensure your code runs smoothly.
—-
Puzzle 0:

**[Insert Puzzle]**
—
Puzzle 1:

**[Insert Puzzle]**
—
Puzzle 2:

**[Insert Puzzle]**
—
**ChatGPT:**

**Prompt for the puzzle generator of ELM and ELM semantic.** This prompt is used for non-autotelic baselines.

---

**ELM and ELM semantic**

**User:** I will give you 3 (Puzzle 0 to Puzzle 2) Python Programming Puzzle (P3). A P3 consists of a problem f and its corresponding solution g. The puzzle is solved if f(g()) == True. Your role is to write 3 new puzzles (Puzzle 3 to Puzzle 5). Note that the first argument of f is the output g(). Make sure to define and set values for all arguments of the function 'f' (excluding the first argument, as it is the solution that needs to be found and given by g).
Both functions, 'f' and 'g' should have matching argument signatures: def f(arg0, arg1=value1, arg2=value2, ...) and def g(arg1=value1, arg2=value2, ...). Please provide all values (value1, value2, ... ) for all arguments. For example f(solution,arg1=1, arg2=2, ...) and g(arg1=1, arg2=2, ...). And you should not use f inside g.
Additionally, make sure to import any necessary libraries to ensure your code runs smoothly.
—-
Puzzle 0:

**[Insert Puzzle]**
—
Puzzle 1:

**[Insert Puzzle]**
—
Here is the puzzle to mutate:
Puzzle 2:

**[Insert Puzzle to mutate]**
—
Could you please mutate the Puzzle 2 into 3 new correct Python Programming Puzzles (from Puzzle 3 to Puzzle 5)? Please, ensure the mutated puzzles are meaningfully different from the existing puzzles.

**ChatGPT:**

---

**Example Generation**

**Puzzle to mutate:**

```python
from typing import*
def f(n: int, lst=['apple', 'banana', 'orange', 'grape']) ->
    bool:
    """Check if the given element n is a prefix of any element
        in the list lst"""
    for word in lst:
        if word.startswith(n):
            return True
    return False

def g(lst=['apple', 'banana', 'orange', 'grape']):
    return lst[1]

assert f(g())
```

**ChatGPT:**

```python
from typing import*
import math
def f(n: int, lst=[1, 2, 3, 4, 5]) -> bool:
    """Check if the given list contains any perfect square
        number and if it is divisible by n."""
    for num in lst:
        if math.isqrt(num)**2 == num:
            return n % num == 0
    return False
def g(lst=[1, 2, 3, 4, 5]):
    for num in lst:
        if math.isqrt(num)**2 == num:
            return num
    return None
assert f(g())
```

## A.3    SAMPLING EXAMPLES FOR CONFUSION MATRIX COMPUTATION

We sample 60 puzzles from one of our ACES experiments that we manually label with the sat of skills. To make sure all skills are represented in the sampled puzzles while ensuring we stay close enough to the distribution of puzzles encountered in training, we sample half 1 of these puzzles uniformly over all generated puzzles in the seed, and we sample half 2 according to the following principle: each time we sample a new puzzle we sample among the list of puzzles involving the currently least represented skill. While examining the dataset (see Figure 7) we noticed a certain number of outlier puzzles which had been labeled with all the skills, and which would be overrepresented in the dataset because they contain all (and thus rare) skills. When inspecting these puzzles we did not find them to be a true combination of all skills. Thus when sampling half 2 (but not half 1) of the manually labeled puzzles we have excluded the anomalous puzzles labeled with all labels.

## A.4    EXAMPLES OF GENERATED PUZZLES

In this section we present a few puzzles and solution generated by our different methods and examine them qualitatively. In example A.4, the generated puzzles combines a string manipulation problem, a grid problem and a recursion problem, through the search for a path in a grid filled with characters of a given input string. The example also illustrates the drift in task semantics mentioned in the main text: in this example the function g only gives the argument of the puzzle, which is both defined and solved in f. We additionally confirm our intuition of the drift in meaning of the puzzles through histograms of the proportion between puzzle and solution complexity, in Figure 8. The intuition is

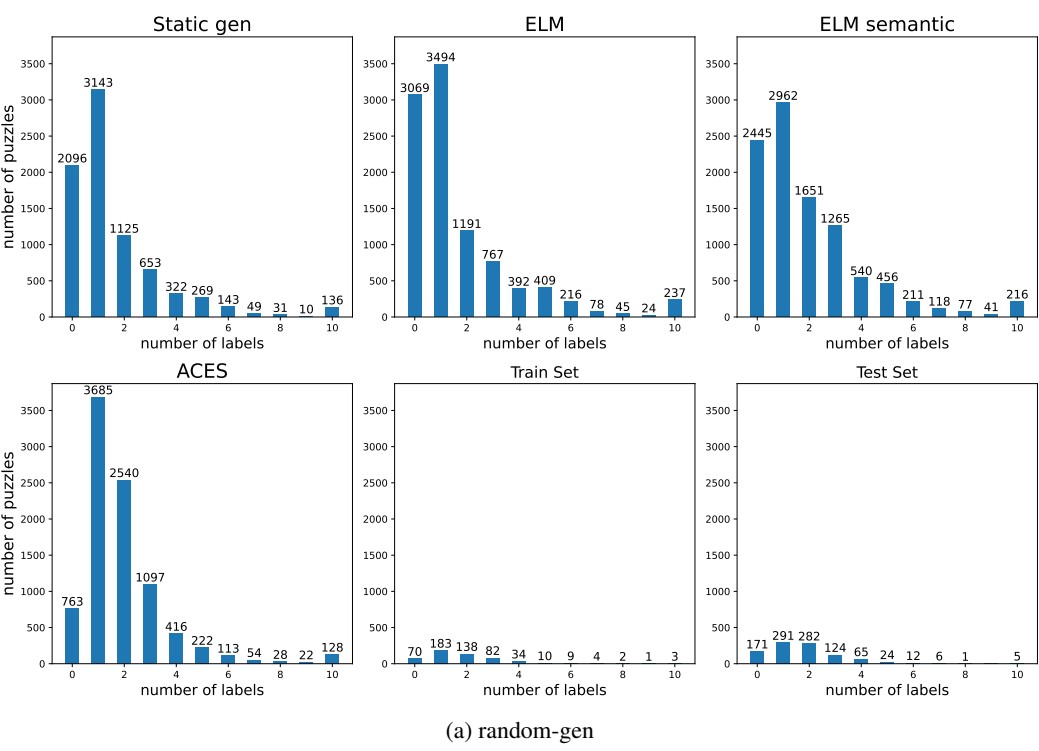

(a) random-gen

Figure 7: Distributions of number of skills labeled by ChatGPT for (a) ACES, (b) ACES-random, (c) ELM-Semantic, (d) ELM, (e) static gen. A noteworthy effect of the goal-targeting in ACES and ACES-random is the low number of puzzle with no skill labels compared to the other methods. Goal-targetting seems to have an effect of how much generated puzzles fit to the predefined ontology.

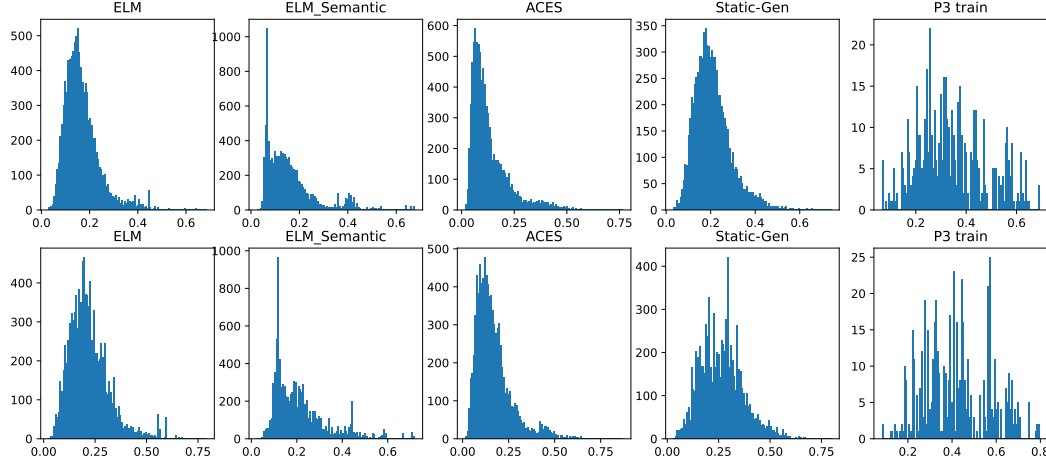

Figure 8: Relative complexity of the generated solutions. We report the number of characters (top) and the size of nodes in the parsed abstract syntax tree (bottom) for the four algorithms and P3's train set (rightmost column). We can see a loss of relative complexity in all methods, less severe in Static-Gen.

that methods that generate puzzles with very short (and thus very simple) solutions have shifted from implementing algorithms in puzzles rather than in solutions. There is a clear difference between the train set from the P3 dataset and the data generation methods, with Static-Gen suffering less. This might have an influence on downstream performance.

**Combination of String and Grid (ACES)**

```python
from typing import*
from typing import List

def f(grid: List[List[int]], s: str) -> bool:
    """
    Given a 2D grid of characters and a string, check if the
        string can be formed by traversing adjacent cells in
        the grid.
    The goal is to find a path in the grid that forms the
        given string.
    """

    def dfs(i: int, j: int, k: int) -> bool:
        # Base case: If we have reached the end of the string,
            return True
        if k == len(s):
            return True

        # Check if the current cell matches the next character
            in the string
        if grid[i][j] == s[k]:
            # Mark the current cell as visited
            visited[i][j] = True

            # Check each neighbor of the current cell
            for di, dj in [(1, 0), (-1, 0), (0, 1), (0, -1)]:
                ni, nj = i + di, j + dj
                # If the neighbor is within the grid
                    boundaries and has not been visited
                if 0 <= ni < len(grid) and 0 <= nj < len(grid
                    [0]) and not visited[ni][nj]:
                    # Recursively search for the remaining
                        characters in the string
                    if dfs(ni, nj, k + 1):
                        return True

            # Mark the current cell as unvisited
            visited[i][j] = False

        return False

    # Start the search from each cell in the grid
    for i in range(len(grid)):
        for j in range(len(grid[0])):
            visited = [[False] * len(grid[0]) for _ in range(
                len(grid))]
            if dfs(i, j, 0):
                return True

    return False

def g():
    grid = [
        ['a', 'b', 'c', 'e'],
        ['s', 'f', 'c', 's'],
        ['a', 'd', 'e', 'e']
    ]
    return (grid, "abcced")
```

**Example generation for ACES**

Label: [1, 0, 0, 0, 0, 0, 0, 0, 0, 0]

```python
from typing import*
from typing import List

def f(nums: List[int]) -> List[int]:
    """Given a list of integers, sort the list in non-
        decreasing order using the bubble sort algorithm.
    Return the sorted list.
    """
    def bubble_sort(nums):
        n = len(nums)
        for i in range(n):
            for j in range(0, n - i - 1):
                if nums[j] > nums[j + 1]:
                    nums[j], nums[j + 1] = nums[j + 1], nums[j
                        ]
        return nums

    return bubble_sort(nums)

def g():
    nums = [9, 5, 2, 8, 1]
    return nums
```

Labels: [1, 1, 0, 0, 0, 1, 0, 0, 0, 0]

```python
from typing import*

def f(s: str, n: int) -> bool:
    """Check if the given string is a palindrome and if the
        given number is divisible by 3 and 5."""
    is_palindrome = s == s[::-1]

    is_divisible_by_three_and_five = n % 3 == 0 and n % 5 == 0

    return is_palindrome and is_divisible_by_three_and_five

def g():
    s = "deed"
    n = 30
    return s, n
```

**Example generation for ELM-Semantic**

Label: [0, 0, 0, 0, 0, 0, 0, 0, 0, 0]

```python
from typing import *

def f(numbers: List[int], target_sum=100) -> bool:
    """Check if there are three numbers in the list that add
       up to the target sum."""
    for i in range(len(numbers)):
        for j in range(i+1, len(numbers)):
            for k in range(j+1, len(numbers)):
                if numbers[i] + numbers[j] + numbers[k] ==
                   target_sum:
                    return True
    return False

def g(target_sum=100):
    return [10, 20, 30, 40, 50, 60, 70, 80, 90]
```

Labels: [0, 1, 0, 0, 0, 1, 0, 0, 0, 0]

```python
from typing import *

def f(words: List[str], prefix='in') -> bool:
    """Check if any word from the list starts with the given
       prefix"""
    return any((w.startswith(prefix)) for w in words)

def g(prefix='in'):
    return ['input', 'information', 'innovation', 'great']
```

**Example generations for ELM**

Labels: [1, 1, 0, 0, 0, 1, 0, 1, 0, 0]

```python
from typing import*
from typing import List

def f(names: List[str], name_length: int) -> bool:
    """Check if there exists a name in the given list that has
        the specified length"""
    for name in names:
        if len(name) == name_length:
            return True
    return False

def g(names=['John', 'Alice', 'Bob', 'Eve'], name_length=5):
    return names, name_length
```

Labels: [0, 0, 0, 0, 0, 0, 0, 0, 0, 0]

```python
from typing import*

def f(arr: List[int], k=10) -> bool:
    """Find if there exists a pair of integers in the array
        whose sum is equal to k"""
    for i in range(len(arr)):
        for j in range(i+1, len(arr)):
            if arr[i] + arr[j] == k:
                return True
    return False

def g(k=10):
    return [1, 2, 3, 4, 5, 6, 7, 8, 9]
```

**Example generations for Static-Gen**

Labels: [1, 0, 0, 1, 0, 0, 0, 0, 0, 0]

```python
from typing import*

def f(n: int, m=5) -> bool:
    """Check if n is divisible by any prime number less than m
        """
    primes = [2, 3, 5, 7, 11, 13, 17, 19, 23, 29, 31, 37, 41,
        43, 47, 53, 59, 61, 67, 71, 73, 79, 83, 89, 97]
    return any(n % prime == 0 for prime in primes if prime < m
        )

def g(m=5):
    return 100
```

Label: [0, 0, 0, 0, 0, 0, 0, 0, 0, 0]

```python
from typing import*

def f(x: int, lst1=[7, 5, 3, 1], lst2=[1, 3, 5, 7]) -> bool:
    """Check if two lists are equal when reversed"""
    return lst1[::-1] == lst2

def g(lst1=[7, 5, 3, 1], lst2=[1, 3, 5, 7]):
    return lst1[::-1]
```

## A.5 ADDITIONAL RESULTS

Additionally, we can visualize language diversity by projecting the semantic embeddings into 2D space via dimensionality reduction techniques like UMAP. We expect to measure diversity in a set of puzzles by looking at the coverage of the map. As the diversity increases, they should achieve both a wider spread of embeddings and potentially clearer cluster separation compared to baselines, showcasing greater linguistic diversity. of puzzles' embedding projected in 2d plane with UMAP.

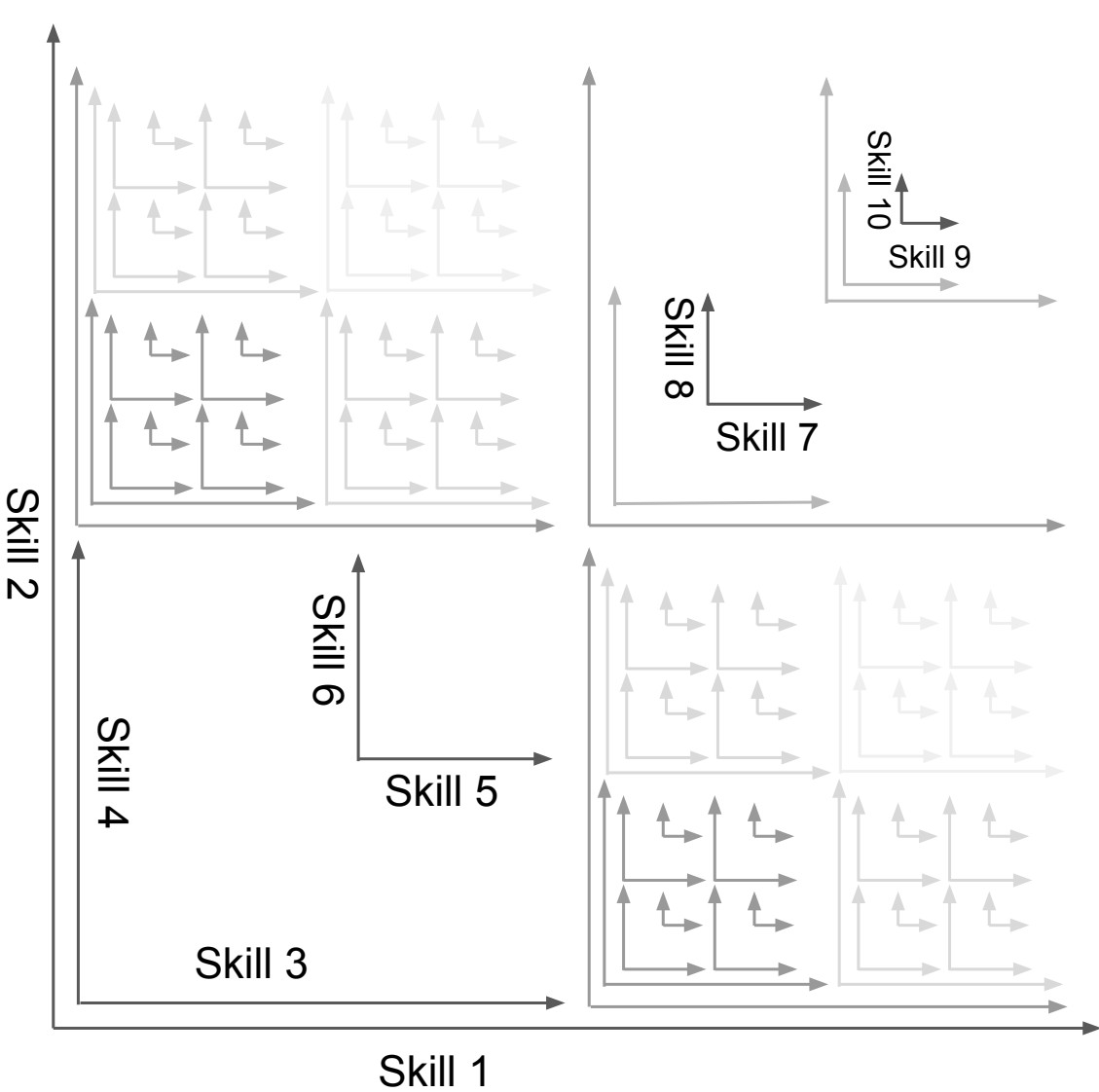

Figure 9: Explanation of the representation used in the Figure 10

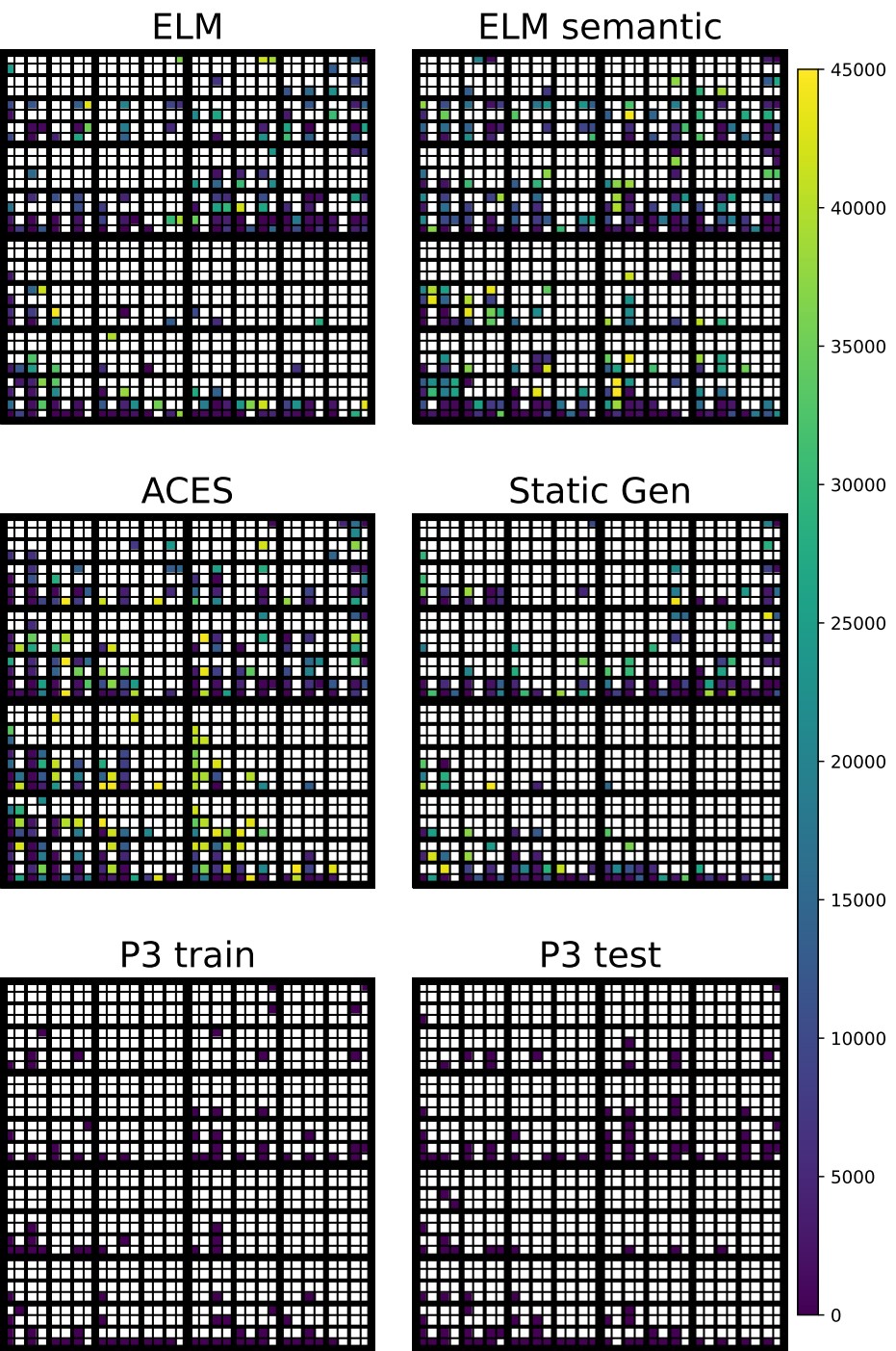

Figure 10: Niche discovered over time 2d representation of cells filled in the skill space. More detail on the representation is given in the Figure 9

Figure 11: UMAP projection of the Code5P-110M embeddings of discovered puzzles for one seed of each algorithm.

Figure 12: UMAP projection of the WizardCoder-1B embeddings of discovered puzzles for one seed of each algorithm.

Figure 13: UMAP projection of the WizardCoder-3B embeddings of discovered puzzles for one seed of each algorithm.

Figure 14: UMAP projection of the WizardCoder-3B embeddings of discovered puzzles for one seed of each algorithm.

