# OpenReview forum: "ACES: Generating Diverse Programming Puzzles with Autotelic Language Models and Semantic Descriptors"
_ICLR.cc/2024/Conference — Submitted to ICLR 2024_

### Official Review · Reviewer_XBPq · 2023-10-21

**Soundness:** 2 fair
**Presentation:** 4 excellent
**Contribution:** 1 poor
**Rating:** 3
**Confidence:** 5

**Summary:**

The paper presented their approaches to generate diverse and solvable programming questions leveraging LLMs.

**Strengths:**

The writing is easy to follow and the entire generation and evaluation process are presented with adequate details.

**Weaknesses:**

The contribution of this paper is pretty poor. There are several severe problems:

1. The observation that LLMs can generate tasks outside the P3 dataset (the training set that is used for prompting), is basically because the LLM has been pre-trained on many larger scale coding question datasets. For example, the combination of string and grid question on page 21 is a canonical programming task. Even if it's not in P3 dataset, it's highly likely that LLM has seen this during pretraining.

2. The semantic descriptor is sort of defined arbitrarily. It's fine if only used to prompt the LLM, but not quite reasonable in measuring interesting-ness and diversity. For example, solely from the human labeled P3 dataset, what's the correlation coefficients between each pairs of labels? As far as the reviewer can see, many labels are highly related, e.g. sorting and searching -> stacks and queues or tree or recursion. Using Hamming distance over those highly correlated field is problematic. Similarly, in the diversity measurement, the grid of labels is reasonable only if categories are almost independent and orthogonal to each other.

3. The process relies on LLM to label the generated tasks and add them into the prompt example pool. View from the Figure 3, this labelling accuracy is far from satisfactory, especially the generation process is iterative and accumulates the errors.

**Questions:**

1. What does the first contribution bullet in the introduction section mean? **We define the notion of semantic descriptors to leverage LLMs for the encoding of high-dimensional textual data into hard-to-compute, abstract and interpretable features of interest**

2. What does the first sentence in Figure 1 caption mean?

3. The page limit of 9 suggests that the reproducibility and ethics should be put after the references.

4. The reviewer agrees that interesting-ness and diversity measurement itself could be a good contribution if they had been well developed.

---

> ### Author Response · Authors · 2023-11-17
> **Answer**
>
> We thank Reviewer XBPq for their detailed and helpful comments and questions.
>
> >“The observation that LLMs can generate tasks outside the P3 dataset (the training set that is used for prompting), is basically because the LLM has been pre-trained on many larger scale coding question datasets. For example, the combination of string and grid question on page 21 is a canonical programming task. Even if it's not in P3 dataset, it's highly likely that LLM has seen this during pretraining.”
>
> This is true for any generative algorithm based on an LLM. Whether puzzles exist somewhere in the world or not, we’re asking whether we can build an algorithm able to generate a diversity of such puzzles along a predefined set of dimensions of interest, and whether this diversity is greater than the one produced by existing algorithms. This is what this paper shows. We make no claim about the historical novelty of the generated problems.
>
> >“The semantic descriptor is sort of defined arbitrarily. It's fine if only used to prompt the LLM, but not quite reasonable in measuring interesting-ness and diversity. For example, solely from the human labeled P3 dataset, what's the correlation coefficients between each pairs of labels? As far as the reviewer can see, many labels are highly related, e.g. sorting and searching -> stacks and queues or tree or recursion. Using Hamming distance over those highly correlated field is problematic. Similarly, in the diversity measurement, the grid of labels is reasonable only if categories are almost independent and orthogonal to each other.”
>
> We agree with the concern about the selection of descriptor features for evaluation and, for this reason, include measures of diversity computed in three embedding spaces that were not used during training. This point is addressed in the general answer as it was raised by other reviewers as well.
>
> We’re not sure to understand the concern about hamming distance and the diversity measurement (entropy over cells, cell count), could you be more specific?
>
>
> > “The process relies on LLM to label the generated tasks and add them into the prompt example pool. View from the Figure 3, this labelling accuracy is far from satisfactory, especially the generation process is iterative and accumulates the errors.”
>
> This point is similar to the one above and is also answered in the general answer. The concern about the accumulation of errors is only valid for the evaluation using semantic descriptors but cannot possibly explain increased diversity in the three embedding spaces. Based on these three latter metrics alone, we can reasonably argue that our method generates a stronger diversity than existing ones. Furthermore, the bias introduced by the imperfect labeling is constant across algorithms (it should impact baselines in the same way).
>
> >“What does the first contribution bullet in the introduction section mean? We define the notion of semantic descriptors to leverage LLMs for the encoding of high-dimensional textual data into hard-to-compute, abstract and interpretable features of interest”
>
> One contribution of our paper is to give experimenters the possibility to define abstract and interpretable features of interest that best capture the dimensions of the type of diversity they want to generate. Whereas before they could only define features they could hand-code a function for, leveraging LLM gives them more expression power.
>
> > “What does the first sentence in Figure 1 caption mean?”
>
> “ACES maintains an archive of discovered puzzles grouped into cells indexed by their semantic representation (skill combination)” means that ACES stores and accumulates valid generated puzzles into an archive organized into different cells, where each cell corresponds to a particular semantic representation. Each representation is a binary vector describing whether skill i is required (1) or not (0) to solve the puzzle.
>
> >“The page limit of 9 suggests that the reproducibility and ethics should be put after the references.”
>
> This has been updated.

---

### Official Review · Reviewer_ufAe · 2023-10-26

**Soundness:** 2 fair
**Presentation:** 3 good
**Contribution:** 1 poor
**Rating:** 3
**Confidence:** 4

**Summary:**

This paper studies the task of automatically generating python programming puzzles.

First, it proposes to use high-dimensional 0-1 vectors (referred to as “semantic descriptors”) to describe different semantic features of programming puzzles, measure the distance between puzzles and diversity of puzzle sets.

Second, it introduces an algorithm ACES that uses these semantic descriptors to generate diverse programming puzzles. Specifically, this algorithm

- randomly samples a descriptor as the goal for generation,
- retrieves similar puzzles according to the hamming distance between the goal descriptor and the descriptor of the known puzzles,
- prompts a language model with the goal descriptor and the retrieved puzzles to generate new puzzles.

The authors evaluate the validity of LLM-labeling for the descriptors and the diversity of ACES-generated puzzles measured by semantic descriptors and other representations. They also examine whether a diversity of generated puzzles can train better code models.

**Strengths:**

1. Automatic generation of programming puzzles (and more broadly, other tasks) is an interesting problem and could be very useful for the purposes the paper mentions.
2. The paper’s evaluation of diversity is multidimensional and extensive. I am convinced that the proposed algorithm does improve the diversity of generated puzzles.
3. The qualitative analysis and the examples in the appendix show that ACES was able to generate some very non-trivial puzzles.

**Weaknesses:**

### 1. There is a lack of discussion on literature related to diverse data generation.

In my opinion, there are important previous studies that are not cited and discussed in your paper. Therefore, I am not sure about the novelty of your algorithm.

As I understand, the proposed ACES algorithm achieves diversity with two approaches:

- Specifying diverse goals with random “semantic descriptors”. In other words, you randomly sample a set of features (like “graph theory”, “optimization algorithm”) and ask the model to make sure the generated puzzles include these features.
- Mutating randomly sampled puzzles from the archive to generate new samples. Since the randomly sampled puzzles are diverse, their mutations will be diverse.

The same two approaches have been extensively used in recent studies on diverse data generation and curation that try to create more data to train and fine-tune language models. Here are some examples.

**Mutation for diversity.** Unnatural Instructions [1] starts with 15 seed examples of (instruction, input, constraints) tuples and mutates a 3-example demonstration to generate new examples. Self-Instruct [2] also does similar things with (instruction, input, output) tuples and it randomly samples instructions from both the initial pool (like your P3 train set) and the generated pool (like your generated puzzle-solution pairs) to start their mutation. Evol-Instruct [3] uses an even more complicated evolutionary algorithm to mutate and generate complex instruction samples.

********************************************************************Goal specification for diversity.******************************************************************** The authors claim that the semantic descriptor (binary vector that indicates which features to include) of ACES is novel and easy to interpret. I agree with the interpretability argument but not the novelty argument. TinyStories [4] also tries to create diverse stories to train language models by creating a list of story features and asking language models to include randomly sampled features from the list each time. Evol-Instruct [3] also involves rewriting and evolving the prompts towards different goals.

More importantly, even in the domain of generating coding tasks, these two approaches (mutation and goal specification) have been used to generate diverse coding tasks (though not programming puzzles). Gunasekar, Suriya, et al. [5], Code Llama [6], and WizardCoder [7] all use these approaches to instruction-tune language models.

I would love to see more discussions on these related works and how ACES is different from (and better than) them.

### 2. ChatGPT labeled semantic descriptor isn’t good enough for measuring semantic diversity.

First, I’m not sure the confusion matrices in Figure 3 support the claim that “puzzle labeler demonstrates high true positive rates” because only 3 among the 10 dimensions have true positive rates over 0.6. Could you be saying it demonstrates high true negative rates? Even so, I’m a bit concerned about the accuracy of the labeler.

Second, I agree with you that “the classification does not need to be perfect to drive diversity.” However, I do think it needs to be more accurate to *******measure******* diversity.

Your entire claim about “diversity in semantic space” is supported by a not-so-accurate labeler of semantic descriptors. Although I think it might be true, the supporting evidence you gave isn’t strong.

### 3. There is a lack of evaluation beyond diversity.

I appreciate your efforts in evaluating diversity with multiple different metrics. However, I think there’s more to data quality than diversity. While the validity check of puzzles `f(g())==True` does say something about the generated puzzles, they could still be meaningless. Therefore, I would love to see more evaluations or arguments about the quality of the generated data.

Specifically, in your qualitative analysis, you mentioned that the generation processes “shift the algorithmic load from g to f, in which case g only provides arguments for f.” I wonder if there were other shifts in the generation processes and if these shifts could lead to degraded puzzles. For example, could f ignore what g is doing or what the algorithmic part in itself is doing and just return True?

Another worry I have is that your goal specification could lead to an infeasible combination of features. Randomly sampled semantic descriptors could involve semantic labels that hardly appear together and create absurd problems.

As demonstrated by your section 4.4, diversity does not entail performance gain. Even though that is the case, I would love to see more evaluation and discussion about the quality of the generated puzzles beyond diversity because quality is important for education, data augmentation, and scientific discovery.

### Reference

[1] Honovich, Or, et al. "Unnatural instructions: Tuning language models with (almost) no human labor." *arXiv preprint arXiv:2212.09689* (2022).

[2] Wang, Yizhong, et al. "Self-instruct: Aligning language model with self generated instructions." *arXiv preprint arXiv:2212.10560* (2022).

[3] Xu, Can, et al. "Wizardlm: Empowering large language models to follow complex instructions." *arXiv preprint arXiv:2304.12244* (2023).

[4] Eldan, Ronen, and Yuanzhi Li. "TinyStories: How Small Can Language Models Be and Still Speak Coherent English?." *arXiv preprint arXiv:2305.07759* (2023).

[5] Gunasekar, Suriya, et al. "Textbooks Are All You Need." *arXiv preprint arXiv:2306.11644* (2023).

[6] Roziere, Baptiste, et al. "Code llama: Open foundation models for code." *arXiv preprint arXiv:2308.12950* (2023).

[7] Luo, Ziyang, et al. "WizardCoder: Empowering Code Large Language Models with Evol-Instruct." *arXiv preprint arXiv:2306.08568* (2023).

**Questions:**

I would appreciate your answers to the questions raised in the weakness section.

---

> ### Author Response · Authors · 2023-11-17
> **Answer (part 1)**
>
> We thank Reviewer ufAe for their detailed and helpful comments and questions.
>
> **Comparison to related works:**
> > “Mutation for diversity. Unnatural Instructions [1] starts with 15 seed examples of (instruction, input, constraints) tuples and mutates a 3-example demonstration to generate new examples. Self-Instruct [2] also does similar things with (instruction, input, output) tuples and it randomly samples instructions from both the initial pool (like your P3 train set) and the generated pool (like your generated puzzle-solution pairs) to start their mutation. Evol-Instruct [3] uses an even more complicated evolutionary algorithm to mutate and generate complex instruction samples.
>
> >Goal specification for diversity. The authors claim that the semantic descriptor (binary vector that indicates which features to include) of ACES is novel and easy to interpret. I agree with the interpretability argument but not the novelty argument. TinyStories [4] also tries to create diverse stories to train language models by creating a list of story features and asking language models to include randomly sampled features from the list each time. Evol-Instruct [3] also involves rewriting and evolving the prompts towards different goals.
>
> >More importantly, even in the domain of generating coding tasks, these two approaches (mutation and goal specification) have been used to generate diverse coding tasks (though not programming puzzles). Gunasekar, Suriya, et al. [5], Code Llama [6], and WizardCoder [7] all use these approaches to instruction-tune language models.”
>
> We thank the reviewer for extensive pointers to relevant prior work and agree additional discussion of code data augmentation methods is needed. We updated the related work section of the manuscript to include this. A general comment here is that these methods, coming from the synthetic training data generation literature, do not aim to optimize for diversity as a goal; usually when they include diversity promoting mechanisms it is for the sake of downstream performance only. As another general note, in addition to the semantic descriptors we introduce, there are two important aspects of our method which are goal selection (which the reviewer points out) and the evolutionary search aspect (which is different from mutation only). All algorithms we evaluate in our experiments except the static-generation are non-stationary since they add their generated data back into the candidates one can sample from. The generative process evolves over time by reusing discoveries for further discoveries. Unnatural instructions simply generates novel instructions from few-shot examples and is neither autotelic, nor diversity-maximizing or non-stationary. Self-Instruct uses both seed examples and generated examples so does display non-stationarity. However it does not maximize diversity, and it is not autotelic. Evol Instruct (as is used both in WizardLM and WizardCoder) is not autotelic, even if it is non-stationary; in any case there is no explicit diversity-maintaining mechanism and additionally there are absolutely no evaluations of the diversity of the produced dataset which prevents any conclusion.
>
> We thank the reviewer for pointing out TinyStories to us, a work we were unaware of that is quite relevant. The proposed goal-generating process is indeed autotelic but it is not evolutionary; the generated stories are never put back in the pool. It is probably totally satisfactory for their domain because they have many different nouns to combine and the requirements to produce a valid story with an LLM are pretty low. The goals they use are a mixture of concrete words (we explicitly want to foster diversity in a space of abstract), and story features such as good or bad endings, which is very much similar to what we do and we find very interesting; they only have one abstract feature per story however (and do not measure the diversity of generated stories with respect to these features, even if we can guess that GPT4 will do fine at capturing them in their generated data). Evol-Instruct, on the other hand, has different prompts with categories of mutations but as mentioned earlier they do not attempt to measure the diversity of the actually generated samples.
>
> (the rest of the answer is in the part 2)

---

> > ### Author Response · Authors · 2023-11-17
> > **Answer (part 2)**
> >
> > In the domain of coding task generation for instruction fine-tuning, no method combines all elements we mentioned earlier: non-stationarity, explicit diversity maximization, and goal-targeting. Evol-Instruct was already mentioned. As far as the authors can see, CodeLlama instruction generation for programming tasks is done in a pretty standard way with none of the features of our method, similar to Unnatural Instructions. (We do find their use of unit tests conditioned on generated language instructions to be very interesting). And finally, while the generation process of the phi-1 paper (Gunasekar et. al.) seems very interesting, we could not find any description of the algorithm used (from the author list it seems as if they might have used some goal targeting like the authors of TinyStories, who are both involved). If you have any additional pointers, or if we missed the point in the paper where this was mentioned we will be glad to discuss it but for now we will consider comparison impossible.
> >
> > > Semantic evaluation metric: “First, I’m not sure the confusion matrices in Figure 3 support the claim that “puzzle labeler demonstrates high true positive rates” because only 3 among the 10 dimensions have true positive rates over 0.6. Could you be saying it demonstrates high true negative rates? Even so, I’m a bit concerned about the accuracy of the labeler. Second, I agree with you that “the classification does not need to be perfect to drive diversity.” However, I do think it needs to be more accurate to measure diversity. Your entire claim about “diversity in semantic space” is supported by a not-so-accurate labeler of semantic descriptors. Although I think it might be true, the supporting evidence you gave isn’t strong.”
> >
> > Thanks for pointing out that sentence, we meant high true negative rates and corrected it. The confusion matrices indeed show that our labeler is not perfectly aligned with human judgements on these dimensions. This said, its low false negative rates limit the possibility for puzzles to be mapped to cells they do not belong to. Having imperfect labeling is not such a problem in the context of diversity production, it only impacts the actual diversity of the generated puzzles and results in the sampling of suboptimal in-context examples. It is a bigger problem when used for evaluation. This point is discussed in our general answer as it was raised by several reviewers. Our main argument is that it still provides some information about the generated diversity (labeling issue is constant across algorithms), especially when coupled with the qualitative study of generated puzzles in Section 4.3. In terms of diversity evaluation, this result is supported and reinforced by the parallel measures of diversity in three different embedding spaces that were not used in the training procedure. These additional metrics alone should bring enough evidence for our claim that our method allows to generate stronger diversity than existing approaches. These points are now made explicit in section 4.1.

---

> > > ### Author Response · Authors · 2023-11-17
> > > **Answer (part 3)**
> > >
> > > > “I appreciate your efforts in evaluating diversity with multiple different metrics. However, I think there’s more to data quality than diversity. While the validity check of puzzles f(g())==True does say something about the generated puzzles, they could still be meaningless. Therefore, I would love to see more evaluations or arguments about the quality of the generated data.
> > >
> > > >"Specifically, in your qualitative analysis, you mentioned that the generation processes “shift the algorithmic load from g to f, in which case g only provides arguments for f.” I wonder if there were other shifts in the generation processes and if these shifts could lead to degraded puzzles. For example, could f ignore what g is doing or what the algorithmic part in itself is doing and just return True?”
> > >
> > >
> > > >“As demonstrated by your section 4.4, diversity does not entail performance gain. Even though that is the case, I would love to see more evaluation and discussion about the quality of the generated puzzles beyond diversity because quality is important for education, data augmentation, and scientific discovery.”
> > >
> > > Evaluation of data quality is probably the main limitation of the proposed approach. We looked at performance on downstream tasks (inconclusive), filtered on validity, and discussed particular examples (Section 4.3) but this cannot be the whole picture. The quality orInterestingness of a puzzle is at least in part subjective. It might be a function of the difficulty of the puzzle for a given programmer, or maybe the number of ways in which the puzzle could be solved. Evaluating such quality computationally thus remains an open-ended challenge (see an attempt using LLMs in OMNI: Open-endedness via Models of human Notions of Interestingness, Zhang et al., 2023). Another way is to involve human evaluators in the loop, which would require a whole new set of study design that is left for future work (e.g. see the discussion on educational application in our answer to reviewer AUHG). An intermediate approach that we could take is to compute measures of difficulty as the percentage of trials that an LLM solves a given problem and measure quality with a combination of intermediate difficulty (neither too easy nor too hard) and diversity of the generated answers (eg computed in embedding space).
> > >
> > > >“Another worry I have is that your goal specification could lead to an infeasible combination of features. Randomly sampled semantic descriptors could involve semantic labels that hardly appear together and create absurd problems.”
> > >
> > > Pursuing goals is a way to steer the exploration of the agent, here by using relevant examples from the archive to achieve empty cells from that same archive. Whether the goal is achievable or not, whether it is achieved or not has only an indirect effect on the quality of the exploration. What matters is that the generated behavior (here the puzzles) are more diverse than if they were generated via another method (see our baseline). Our results show that this is the case. Of course, goal sampling may matter, and goal curriculum learning studies have shown that focusing on goals of intermediate difficulty, novel goals or goals where the agent progresses the most might bring significant advantages in some domains. Because here it is hard to know which goals are achievable and which are not, we simply focus on sampling goals that we expect to be of intermediate difficulty here: goals that are not already reached in the archive but are not too far from goals we’ve reached in the past.
> > >
> > > In interesting cases, it could be that the experimenter does not see any generation that could fit a particular cell a priori, but gets surprised as the diversity search finds one. Eventually, we’d like to observe such behavior where the diversity producing algorithm actually surprises us by its generations.

---

> > > > ### Comment · Reviewer_ufAe · 2023-11-20
> > > >
> > > > Thank you for the response. It has addressed some of my concerns. I'm more convinced about ACES being able to actually improve the diversity of generated puzzles.
> > > >
> > > > However, my primary concern about this paper remains: what does it offer beyond diversity?
> > > >
> > > > In terms of novelty, other than combining the features of existing works, the main novelty claimed by the authors in the response is "optimizing diversity as a goal". I would be ok with this novelty as long as it gets us more than diversity.
> > > >
> > > > As the authors have admitted themselves in the response:
> > > >
> > > > > Evaluation of data quality is probably the main limitation of the proposed approach
> > > >
> > > > This contradicts what the authors proposed in the introduction regarding applications:
> > > >
> > > > > education (generating problems for students to solve), data augmentation (generating problems and solutions for AI model training), or automated scientific discoveries (e.g. discovering new scientific problems and their solutions)
> > > >
> > > > I would really appreciate seeing examples or quantitative results demonstrating how ACES can serve these purposes (or other similar purposes).
> > > >
> > > > I think the idea mentioned in the last part of the response about discovering surprising combinations could be intriguing:
> > > >
> > > > > it could be that the experimenter does not see any generation that could fit a particular cell a priori, but gets surprised as the diversity search finds one. Eventually, we’d like to observe such behavior where the diversity producing algorithm actually surprises us by its generations.
> > > >
> > > > But is there any concrete example that the authors can provide to convince me that this is indeed the case?
> > > >
> > > > I would consider raising my rating if I can be convinced that ACES truly contributes more than diversity.

---

> ### Author Response · Authors · 2023-11-23
>
> >>This contradicts what the authors proposed in the introduction regarding applications:
>
> >education (generating problems for students to solve), data augmentation (generating problems and solutions for AI model training), or automated scientific discoveries (e.g. discovering new scientific problems and their solutions)
>
>
> Thank you for your feedback. While we could have devoted more effort to evaluating the data quality, this does not necessarily conflict with the proposed applications, particularly in education. For more details, see 'Answer (part 4)' in our response to reviewer AUHG (https://openreview.net/forum?id=hqUznsPMLn&noteId=6FIypHHOgY).
>
> >But is there any concrete example that the authors can provide to convince me that this is indeed the case?
>
> In our experiment, we utilized an LLM proficient in programming, though not the most advanced in recent coding model releases. Despite this, our goal-directed methodology enabled the LLM to create creative and novel coding puzzles not found in archives from other baselines. For instance, consider this example puzzle (docstring of the puzzle): "Given a grid of 0s and 1s, determine if there exists a path from the top-left corner to the bottom-right corner, where you can only move down, right, or diagonally down-right. Additionally, the sum of the numbers along the path must be a prime number."
>
> While this combination might not be overly surprising to an expert, it is nonetheless non-trivial. It shows the potential of our approach. Furthermore, it's worth noting that employing a more advanced LLM could potentially lead to even more creative and surprising combinations. This suggests that as AI models continue to evolve, the scope for generating intricate and intellectually stimulating puzzles in educational settings may expand significantly, offering further evidence of the practical applications of our proposed methods.

---

### Official Review · Reviewer_AUHG · 2023-10-31

**Soundness:** 3 good
**Presentation:** 3 good
**Contribution:** 2 fair
**Rating:** 5
**Confidence:** 4

**Summary:**

The paper introduces ACES (autotelic code exploration with semantic descriptors), a prompting-based algorithm that uses an LLM (ChatGPT) to produce a diverse set of programming puzzles. The algorithm first uses the LLM to label an existing base set of puzzles (P3, the Python Programming Puzzles dataset) as to which of ten skills (aka semantic descriptors) they require. It then proceeds iteratively to generate new puzzles. To generate each new puzzle it uses a nearest neighbor approach to construct a few-shot prompt using existing puzzles, aimed at generating a puzzle requiring a particular subset of the ten skills. It uses this prompt with an LLM to generate a puzzle-solution pair, which it keeps if the puzzle-solution pair passes. The resulting set of puzzles exhibits high diversity both according to the semantic descriptors and according to an embedding based measure of diversity. Despite the diversity of the puzzles produced, fine-tuning using the produced puzzle set does not help on the test puzzle set compared with fine-tuning on a less diverse set of puzzles produced by a baseline approach.

**Strengths:**

* The approach is quite simple, and given access to an instruction-following LLM like ChatGPT, it would be straightforward to reproduce a comparable approach to ACES. The prompts included in the Appendix and the algorithm provided in the text aid in this reproducibility meaningfully. (The reliance on gpt-3.5-turbo-0613 unfortunately time-limit precise reproducibility, but other models will slot into the algorithm without issue even once gpt-3.5-turbo-0613 is no longer available.)
* The approach leads to greater diversity in programming puzzles compared to the static gen, ELM semantic, and ELM baselines. The diversity is evident both in the number of cells covered (the objective the algorithm is designed for) and through an embedding based similarity metric.
* The qualitative inspection of the samples generated by the ACES approach is welcome and informative. Thank you for including this.
* The approach seems readily generalizable from the 10 selected semantic descriptors for programming puzzles to other domains where characteristics of examples can be represented as a set of features or properties present/not-present.

**Weaknesses:**

* ACES relies on a manually crafted set of semantic descriptors that is specific to the task of generating programming puzzles. The paper does not explore the role that the selection of these semantic descriptors plays on the resulting set of programming puzzles, and does not provide insight into how the approach works in non-programming puzzle domains where these pre-selected semantic descriptors would not be appropriate.
  * The semantic descriptor based metric is tailored specifically for the semantic descriptors selected for the approach. If the set of semantic descriptors used in the approach were to change, it's not clear the current semantic descriptor approach (holding the set of semantic descriptors fixed to allow comparing across the change) to measuring diversity would remain meaningful.
* One weakness of this manually crafted set of semantic descriptors is that it places a kind of constraint on the diversity that the approach can produce, and it shifts the burden of identifying this diversity from the ACES algorithm itself to the human seeding the algorithm with semantic descriptors.
* The 2**10 size grid of sets of semantic descriptors is (a) in my opinion quite small. P3 has 636 train puzzles spanning ~60-80 cells, but there are only 1024 cells in total to try to cover (using 45000 generated programs to do so). (b) I expect that many of these cells aren't meaningful, e.g. it doesn't seem that important to identify a programming puzzle that is simultaneously a Sorting and Searching, Counting and Combinators, Tree and Graph, Bit manipulation, string manipulation, recursion, and dynamic programming problem (7 semantic descriptors), and a full 38% of the cells represent the goal of having 6 or more semantic descriptors. Indeed Figure 7 confirms generating programming puzzles with 6+ semantic descriptors is quite rare, yet the algorithm spends significant time trying to do so.
* The primary measure of diversity (number of cells covered in the archive) is both an imperfect measure (the labels used to determine which cell is covered are determined by ChatGPT) and also the same model producing the puzzles (where generation is conditioned on the desired label) is the one predicting the label. Since puzzle generation is conditioned on the goal label and the same model then predicts those labels, there is real risk of the model overestimating the amount of diversity produced. As a toy example of how this overestimation could occur, we could imagine the model adds a comment to each generated puzzle saying what labels that puzzle should have. Then during labeling, the model ignores the code and simply reports the labels indicated by the comment. If this were to occur, the semantic descriptor metric would report all cells get covered (despite not actually achieving the desired diversity). I trust that this precise mechanism is not taking place here, but something directionally similar could easily evade notice and I don't think is being checked or controlled for.

* And of course there is the significant weakness, readily acknowledged and discussed by the paper, that the diversity of puzzles identified by ACES did not help in fine-tuning for the downstream task of puzzle solving on the test set compared with the static gen baseline. I applaud the frank discussion of this weakness in Section 4.4 and the discussion of the paper.

**Questions:**

# Questions and Suggestions

You present two key use cases for generating diverse programs: education and LLM training. You evaluate the latter and find that the diversity provided by ACES is not helpful for the downstream puzzles test task you evaluate on. I am curious to get your thoughts on the former, which you do not evaluate. What properties of diversity do you think are important for the education use case, and how do you think these would be the same/different from the properties of diversity that would lead to positive effects of fine-tuning on a downstream task (like the puzzle test set you work with)?

Why do you select the grid of 2^10 sets of semantic descriptors as your set of possible goals to induce diversity? I ask this question with the following thoughts in mind. First, it seems like many of the cells in this grid don't actually represent diversity of interest (e.g. 38% of cells have 6 or more semantic descriptors which seems an unwieldy amount). Second, it seems like there are more dimensions of diversity not being considered. I brainstorm a few here to make the point: complexity/difficulty, domain, number of inputs, reliance on data dependencies, libraries used, and wall clock runtime.

I am curious to hear any observations you may have made about the embedding based similarity measure. Does its measure correspond to your intuitive sense of similarity? Did you notice any qualitative differences across the three embedding models that you used?

In Section 3.2 you introduce a notion of interestingness, saying that functions R will map all uninteresting samples to the same point. This R notation is then never used, and the notion of interestingness is not further explored. What is meant by interesting, and how do the two representation functions (cosine distance and semantic descriptors) send uninteresting examples to the same point?

In "What's new?" you claim ACES is the first algorithm to use an autotelic LLM to generate diverse artifacts via in-context learning. In seeking to evaluate this claim, I am reminded first of EvoPrompting: Language Models for Code-Level Neural Architecture Search
 (https://arxiv.org/abs/2302.14838) (which is not autotelic, instead optimizing an objective) and I subsequently find concurrent work LLMatic: Neural Architecture Search via Large Language Models and Quality-Diversity Optimization (https://arxiv.org/abs/2306.01102). This concurrent work is also not autotelic. Another such concurrent work is Quality-Diversity through AI Feedback (https://arxiv.org/abs/2310.13032). Finally, does not Colas 2023 Augmenting Autotelic Agents with Large Language Models (https://arxiv.org/abs/2305.12487) also have all of these properties?

I have collected some typographic suggestions for you below. These issues did not meaningfully harm the readability of the paper.

Typo: Figure 1 caption "an archive of dcreate figure from tcolorboxiscovered" is a typo.
Typo: Section 3.2 "the example of 2" -> "the example in Figure 2"
Typo: Section 3.2 "Counting and Combinatoris" -> "Counting and Combinatorics"
Notation: Section 3.2 "k \in [1 : 10]" -> "k \in [1..10]" or "k \in [1, 10]"
Formatting: In Algorithm 1, consider consistent formatting for "LLM" across lines 5 and 8
Typo: Section 3.3 Puzzle labeler. "the generate puzzle" -> "the generated puzzle"
Typo: Section 4.2 Figure -> Figures
Typo: Section 4.2 "more cell" -> "more cells"

---

> ### Author Response · Authors · 2023-11-17
> **Answer (part 1)**
>
> We thank Reviewer AUHG for their detailed and helpful comments and questions.
>
> There were several questions and comments about the choice of semantic descriptors and its impact on the generated diversity:
>
> > “ACES relies on a manually crafted set of semantic descriptors that is specific to the task of generating programming puzzles. The paper does not explore the role that the selection of these semantic descriptors plays on the resulting set of programming puzzles, and does not provide insight into how the approach works in non-programming puzzle domains where these pre-selected semantic descriptors would not be appropriate.”
>
>
> > “One weakness of this manually crafted set of semantic descriptors is that it places a kind of constraint on the diversity that the approach can produce, and it shifts the burden of identifying this diversity from the ACES algorithm itself to the human seeding the algorithm with semantic descriptors.”
>
>
> We answer these in the general answer as these questions came up for other reviewers as well.
>
> > “The semantic descriptor based metric is tailored specifically for the semantic descriptors selected for the approach. If the set of semantic descriptors used in the approach were to change, it's not clear the current semantic descriptor approach (holding the set of semantic descriptors fixed to allow comparing across the change) to measuring diversity would remain meaningful.”
>
> We are not sure to understand this comment, do you point at the fact that the metric we optimize for it is the same we use for evaluation? If so, note that we evaluate the competing methods with diversity metrics computed in three distinct embedding spaces as well (see General Answer).
>
> > “Why do you select the grid of 2^10 sets of semantic descriptors as your set of possible goals to induce diversity? I ask this question with the following thoughts in mind. First, it seems like many of the cells in this grid don't actually represent diversity of interest (e.g. 38% of cells have 6 or more semantic descriptors which seems an unwieldy amount). Second, it seems like there are more dimensions of diversity not being considered. I brainstorm a few here to make the point: complexity/difficulty, domain, number of inputs, reliance on data dependencies, libraries used, and wall clock runtime.”
>
> As discussed in the general answer, we can always think of other representation features and their choice will influence the produced diversity. The definition of the representation function is necessarily subjective because it is precisely the means through which experimenters define what type of diversity they care about. The suggested representation features are all valid and adding them would result in the generation of a different type of diversity along these dimensions as well. We made the choice of not including them but could have.
> Complexity and difficulty are definitely good candidates, especially in educational contexts (see our comment about education below). Including it would require an extra measure of complexity, e.g. by asking GPT or another LLM to solve the problem several times and using the negative success rate as a measure of difficulty. Note that we use a binary measure of difficulty (valid vs invalid) for filtering puzzles that enter the archive. Including the number of inputs or wall clock time may not lead to the production of interesting diversity (e.g. adding time.sleep() instructions, or useless inputs).
> This point is interesting because it illustrates a contribution of our approach. The definition and use of semantic descriptors, by making explicit the space in which the diversity is produced, allows us to discuss the inclusion/exclusion and careful definition of the dimensions of interest. We can thus debate about whether sets A or B of semantic descriptors would allow us to generate the kind of diversity we care the most about for application X. This is not possible with embedding representation functions which, by making these choices implicit, restrict our ability to discuss and debate them. We answer the concern about parts of the representation space being hard or impossible to reach in the following answer.

---

> > ### Comment · Reviewer_AUHG · 2023-11-22
> > **Explanation of observation about the semantic descriptor metric of diversity**
> >
> > >> “The semantic descriptor based metric is tailored specifically for the semantic descriptors selected for the approach. If the set of semantic descriptors used in the approach were to change, it's not clear the current semantic descriptor approach (holding the set of semantic descriptors fixed to allow comparing across the change) to measuring diversity would remain meaningful.”
> > >
> > > We are not sure to understand this comment, do you point at the fact that the metric we optimize for it is the same we use for evaluation? If so, note that we evaluate the competing methods with diversity metrics computed in three distinct embedding spaces as well (see General Answer).
> >
> > My observation is that the semantic descriptor metric of diversity is not a metric that would allow you to fairly compare against very different methods of producing diversity. If you were to try a very different approach (say, a new clever sampling algorithm that is not conditioned on descriptors at all), then even if that new approach were much more effective at producing diversity, it is quite likely that ACES would score higher on the metric. This is because the metric and ACES both use the same notion of diversity and the same descriptors. Even the effect of a small change to ACES, such as increasing the number of descriptors and leaving everything else unchanged, cannot be meaningfully measured with the semantic-descriptor metric. I think this is a key challenge and a real weakness of the metric. I acknowledge that you are also using additional measures of diversity other than this one metric, which is good. My comment was specifically about this one metric.

---

> > > ### Comment · Reviewer_AUHG · 2023-11-22
> > > **Thank you for your responses**
> > >
> > > Thank you for your response to my review.
> > >
> > > I appreciate the exploration of the education context that you gave in Part 4. I can definitely imagine a system like ACES being useful for generating educational content in the future.
> > >
> > > >> “And of course there is the significant weakness, readily acknowledged and discussed by the paper, that the diversity of puzzles identified by ACES did not help in fine-tuning for the downstream task of puzzle solving on the test set compared with the static gen baseline. I applaud the frank discussion of this weakness in Section 4.4 and the discussion of the paper.”
> > > >
> > > > This result should not be seen as a weakness per se.
> > >
> > > Respectfully, I think one of the bigger limitations of the paper is that it does not make a case for the usefulness of the type of diversity it produces. As a curious person and a lover of puzzles myself, I can appreciate the intrinsic value of exploration. And I can certainly see the appeal of a method like this for the education settings you describe. But making a case that the diversity measurements you're making actually reflect useful diversity, for some definition of useful, in my opinion needs to be part of the picture. As is, for the application you actually consider, the particular kind of diversity your algorithm induces was unfortunately not helpful. At the very least it would be important to understand why. (And I expect such an investigation would lead immediately to new ideas for producing different forms of diversity.)
> > >
> > > >> You claim ACES is the first algorithm to use an autotelic LLM to generate diverse artifacts via in-context learning. In seeking to evaluate this claim [...]
> > > >> Finally, does not Colas 2023 Augmenting Autotelic Agents with Large Language Models (https://arxiv.org/abs/2305.12487) also have all of these properties?”
> > > >
> > > > And finally, the work of Colas et. al. does not maximize for any form of diversity and, let alone define its diversity through AI feedback.
> > >
> > > Small note: the claim I was seeking to evaluate was "ACES is the first algorithm to use an autotelic LLM to generate diverse artifacts via in-context learning", not that Colas et. al and your work don't differ in meaningful ways; they most certainly do! Maximizing for diversity is not a prerequisite for producing diverse samples though, and so I think your claim as is might not be literally correct (though with a small tweak could be made true.)

---

> ### Author Response · Authors · 2023-11-17
> **Answer (part 2)**
>
> > “The 2**10 size grid of sets of semantic descriptors is (a) in my opinion quite small. P3 has 636 train puzzles spanning ~60-80 cells, but there are only 1024 cells in total to try to cover (using 45000 generated programs to do so). (b) I expect that many of these cells aren't meaningful, e.g. it doesn't seem that important to identify a programming puzzle that is simultaneously a Sorting and Searching, Counting and Combinators, Tree and Graph, Bit manipulation, string manipulation, recursion, and dynamic programming problem (7 semantic descriptors), and a full 38% of the cells represent the goal of having 6 or more semantic descriptors. Indeed Figure 7 confirms generating programming puzzles with 6+ semantic descriptors is quite rare, yet the algorithm spends significant time trying to do so.”
>
> In the most interesting cases, it may be hard to imagine solutions that would end up in some of the cells our semantic descriptors define. This is fine. We invent many descriptors for movies, music or poetry, it doesn’t mean that we expect specific movies, songs or poems to have any arbitrary conjunction of these features. What’s important is to ‘spread out’ interesting objects by defining axes of variation we care about while ‘conflating’ irrelevant objects together. Assuming a perfect feature description function, if a cell cannot be reached it will never be, but maybe it will be, and then we will have learned something new. There exists a paper full of anecdotes from evolutionary search where the system found creative ways to reach points of a representation space that was thought to be impossible to reach (e.g. a robot found a way to walk without having either of its feet touch the floor by crawling on its back), see The Surprising Creativity of Digital Evolution: A Collection of Anecdotes from the Evolutionary Computation and Artificial Life Research Communities (Lehman et al., 2018).
>
>
> > “The primary measure of diversity (number of cells covered in the archive) is both an imperfect measure (the labels used to determine which cell is covered are determined by ChatGPT) and also the same model producing the puzzles (where generation is conditioned on the desired label) is the one predicting the label. Since puzzle generation is conditioned on the goal label and the same model then predicts those labels, there is real risk of the model overestimating the amount of diversity produced. As a toy example of how this overestimation could occur, we could imagine the model adds a comment to each generated puzzle saying what labels that puzzle should have. Then during labeling, the model ignores the code and simply reports the labels indicated by the comment. If this were to occur, the semantic descriptor metric would report all cells get covered (despite not actually achieving the desired diversity). I trust that this precise mechanism is not taking place here, but something directionally similar could easily evade notice and I don't think is being checked or controlled for.”
>
> This is a fair argument. The current evaluation metric based on semantic descriptors isn’t very reliable, and although this is fine for the purpose of guiding the diversity search, it poses problems when used for evaluation. This is precisely for this reason that we propose other diversity metrics computed from three different embedding spaces that were not used in the training procedure. On its own, the fact that our method outperforms others on these unbiased evaluation metrics should be sufficient to make the argument that our method allows to generate more diversity than existing ones. The semantic descriptor evaluation, in addition to the exploration of generated puzzles, brings additional elements to the study of the generated diversity, although it is not a definitive argument as such for the reason mentioned above. The reviewer is right to point that this evaluation could be in principle hijacked by the puzzle generator. Although we did not implement any mechanism to prevent this from happening, we can have a look at the difference of performance between ELM semantic and ELM in embedding space. These two do not aim for semantic goals, which means that the generator is unaware of the labeling procedure. Therefore, the noise in the semantic diversity metric is the same for both algorithms. Yet, ELM semantic still outperforms ELM in terms of semantic diversity.

---

> ### Author Response · Authors · 2023-11-17
> **Answer (part 3)**
>
> > “And of course there is the significant weakness, readily acknowledged and discussed by the paper, that the diversity of puzzles identified by ACES did not help in fine-tuning for the downstream task of puzzle solving on the test set compared with the static gen baseline. I applaud the frank discussion of this weakness in Section 4.4 and the discussion of the paper.”
>
> This result should not be seen as a weakness per se. As discussed in the introduction and the discussion, data augmentation is only one of several possible motivations for diversity-producing algorithms. Other possible motivations include: the generation of diverse problems for training and evaluating human learners, scientific discovery (eg automatic theorem proving), but also artistic projects (eg generating an interesting diversity of visual artefacts, where the notion of interestingness can be highly subjective: eg beautiful from the point of view of the artist). Our paper only studies quality with respect to the data-augmentation application as it is the one that is easiest to automate. Studying quality for educational or artistic purposes would require extensive and careful human studies, which could be the subject of a stand-alone paper. In general, automating measures of quality or interestingness remain a central challenge in AI (e.g. see an attempt in OMNI: Open-endedness via Models of human Notions of Interestingness, Zhang et al., 2023)

---

> > ### Author Response · Authors · 2023-11-17
> > **Answer (part 4)**
> >
> > > “You present two key use cases for generating diverse programs: education and LLM training. You evaluate the latter and find that the diversity provided by ACES is not helpful for the downstream puzzles test task you evaluate on. I am curious to get your thoughts on the former, which you do not evaluate. What properties of diversity do you think are important for the education use case, and how do you think these would be the same/different from the properties of diversity that would lead to positive effects of fine-tuning on a downstream task (like the puzzle test set you work with)?”
> >
> > Thank you for asking further details on the topic of potential applications of this work in the domain of education. Our lab is indeed actually working on various educational projects leveraging AI techniques, and while the present paper is still at a fundamental research stage, we already envisage a future project where the ideas are reused and evaluated in field studies in an actual educational context.
> >
> > We actually have two educational contexts in mind. The first one consists in helping computer science teachers (at all levels) to design novel, diverse and relevant programming exercises, especially for tests. When teachers design tests on a particular set of programming topics, they need to provide tests that are different for every class and every year (otherwise students could rote learn previous tests): this is extremely time consuming. They might be helped by existing repositories of programming exercises on the internet, but this has two limits: 1) these repositories might not fit very well the peculiar topics they want to teach in their classes; 2) these repositories are often limited in size and diversity.
> >
> > Thus, we believe the ACES approach could be the basis of a system where teachers provide a set of semantic descriptors that match the topics they want to teach, as well as a few examples for some combination of descriptors, and generate diverse exercises for each category. This actually justifies the relevance to consider in our paper human-defined semantic descriptors (as this corresponds to a real world use).
> >
> > However, there are challenges and open questions that would need to be addressed before getting to this educational use case. First, we would need to complement ACES with features measuring the difficulty of exercises, enabling us to target a diversity of difficulties in addition to a diversity of topics involved in the exercises. This may be done e.g. sampling N times solutions from a particular LLMs, and counting the proportion of actual solutions. However, this is an open question to study how this kind of measure of difficulty would match with the difficulty as perceived by human learners. Second, one would need to assess the pedagogical relevance of the exercises, both from the perspective of students and teachers (and it would be very interesting to study how the human measure of pedagogical relevance correlates with usefulness in using the exercises for finetuning and improving performance of LLMs). Feedback on this dimension could also constitute a measure to optimize in an extension of ACES.
> >
> > Another educational context we have in mind is automatic generation of programming exercises in intelligent tutoring systems. Indeed, such edTech systems automatically generate sequences of exercises on topics personalized to particular users (based either on their levels or on their preferences), as well as rehearsal programs. In this context, this is also particularly important to be able to generate continuously new and diverse exercises that fit the topics a learner aims to practice.
> >
> > > “I am curious to hear any observations you may have made about the embedding based similarity measure. Does its measure correspond to your intuitive sense of similarity? Did you notice any qualitative differences across the three embedding models that you used?”
> >
> > We analyze the k-nearest neighbors of various puzzles across three distinct embedding spaces to discern the types of similarities these models extract. Our observations suggest that these models generally align with intuitive understandings of similarity. This means that two programming puzzles that look similar tend to have closely related embedding vectors. For instance, when examining the 10 closest neighbors of one puzzle, we can clearly identify other puzzles that are almost identical. However, it seems that WizardCoder 1b and 3B seem to be slightly better at handling nuances like different input formats. For example, they effectively distinguish between puzzles where a graph or grid is represented as a string of characters and those where it's composed of letters.
> >
> > More broadly, this raises an important question for embedding models. While there is a benchmark for embedding models on textual data, known as the Massive Text Embedding Benchmark (MTEB) Leaderboard, it does not yet include any task based on code datasets.

---

> ### Author Response · Authors · 2023-11-17
> **Answer (part 5)**
>
> > “In Section 3.2 you introduce a notion of interestingness, saying that functions R will map all uninteresting samples to the same point. This R notation is then never used, and the notion of interestingness is not further explored. What is meant by interesting, and how do the two representation functions (cosine distance and semantic descriptors) send uninteresting examples to the same point?”
>
> R is the ‘representation function’ mapping a generated object (here a programming puzzle) into a given representation space (here the embeddings, or the semantic descriptors). In the case of semantic descriptors, R “spreads out” programming puzzles we care about: the ones that are valid and that require various programming skills to get solved and “conflates” uninteresting ones: invalid puzzles are not even represented, and all puzzles that require no skill get mapped to the same unique cell (coordinates [0]^10). Conducting the diversity search in the semantic space is thus expected to be more efficient than conducting it in the original representation space (eg space of token sequences), or even in the embedding space which may not have such properties (ie all puzzles that require no skill to be solved may be mapped to different areas of the embedding space and thus be judged novel). As discussed in the general answer, the choice of the representation function directly influences the produced diversity precisely for this reason.
>
> > “In "What's new?" you claim ACES is the first algorithm to use an autotelic LLM to generate diverse artifacts via in-context learning. In seeking to evaluate this claim, I am reminded first of EvoPrompting: Language Models for Code-Level Neural Architecture Search (https://arxiv.org/abs/2302.14838) (which is not autotelic, instead optimizing an objective) and I subsequently find concurrent work LLMatic: Neural Architecture Search via Large Language Models and Quality-Diversity Optimization (https://arxiv.org/abs/2306.01102). This concurrent work is also not autotelic. Another such concurrent work is Quality-Diversity through AI Feedback (https://arxiv.org/abs/2310.13032). Finally, does not Colas 2023 Augmenting Autotelic Agents with Large Language Models (https://arxiv.org/abs/2305.12487) also have all of these properties?”
>
> You are right to point out that other works have begun to consider LLMs as mutation operators in evolutionary and related algorithms. EvoPrompting is an example of this where the authors mutate the Python code for designing neural architectures and use the negative generalization error as fitness. However, that method does not perform any diversity optimization. LLMatic follows this path by applying a QD algorithm powered by LLM-based generation. Their approach is not autotelic as you have mentioned, but more importantly, the archive they use for their Map-Elites implementation is based on very simple surface-level descriptors (FLOPS, width-to-depth), and does not capture the human-relevant diversity we seek to achieve in this work. QDAIF is the work closest in spirit to ours, even if they perform their experiments in poem and movie review domains that are quite different from the coding domain, mainly because the solvability of programming puzzles can be done by a python interpreter. Note that both LLMatic and QDAIF are under review at this same conference and should not be considered prior work. And finally, the work of Colas et. al. does not maximize for any form of diversity and, let alone define its diversity through AI feedback.
>
> We have updated the appendix to include an "Additional Related Work" section to include a discussion of these works and the crucial aspects in which they differ from ours to make the novelty of our contribution easier to evaluate.
>
> >“I have collected some typographic suggestions for you below. These issues did not meaningfully harm the readability of the paper.”
>
> Thank you, these were all fixed.

---

### Author Response · Authors · 2023-11-17
**General Answer (part 1/2)**

We thank all reviewers for their thorough feedback and suggestions for improving the paper. They all noted the importance of diversity-producing algorithms and the simplicity of our method to achieve stronger, controllable diversity. We noted two main axes of concern shared by the reviewers:
- 1) the choice of semantic descriptors and its impact on diversity,
- 2) the difficulty in evaluating semantic diversity caused by our imperfect puzzle labeler,

We address these two points in this general answer and reviewer-specific comments and questions in their respective answers. The new version of the paper comes with extended discussions addressing the concerns expressed by the reviewers (see pointers in each of the answers).

## 1) About the choice of semantic descriptors:

Diversity-producing algorithms necessarily optimize for diversity in a subjective representation space. For this reason, there is no single ‘good’ representation space in which one should optimize for diversity (Etcheverry et al., 2020). Whether we optimize for diversity within the proposed semantic descriptors space or within a pretrained embedding space, the choice of the representation will necessarily constrain and bias the produced diversity, e.g. two different embedding spaces lead to two different types of diversity.

In the case of embedding spaces, this choice is implicit: we do not really know what kind of features embedding representations capture, and it remains difficult to interpret the resulting diversity. With semantic descriptors, on the other hand, we make this choice explicit: the experimenter expresses a set of features they are interested in and want to produce diversity for. This is especially useful in domains such as education or arts, see our comment on education applications in our answer to reviewer AUHG.

Embedding spaces might be more general and capture nuances that semantic descriptors don’t, but they might also measure novelty that the experimenter may not perceive or care about (e.g. surface details of programming puzzles using different variable names for the same problem, or distances between non programming texts). Semantic descriptors therefore allow the experimenter to explicitly define the axes of variation they are interested in and, compared to previous works, allow them to do so as a high-level of abstraction by off-loading the hard task of defining feature descriptors to an LLM. An interesting result of our paper is that, even though we optimize for diversity in a subjective and fundamentally limited semantic space, we still end up generating more diversity than other methods in unrelated embedding spaces. Section 3.2 now presents an extended discussion of these points.

As experimenters interested in generating a diversity of programming puzzles in the context of this paper, we defined a set of features taken from the outline of a standard computer science textbook (Introduction to Algorithms, H. Cormen et al., MIT Press 2009). These features reasonably capture the axes of variations of the puzzles from P3’s training set. We’ve updated Section 3.2 to better reflect this thought process.

To generate a diversity of movie scripts, we might define semantic descriptors evaluating features like suspense, romance, action, etc. Note that the semantic features do not have to be binary, they could also be measured on a Likert scale, or continuous. No matter the application domain, diversity generation requires the implicit or explicit definition of a representation function that necessarily both restricts but also sustains the produced diversity.

To summarize, we agree that the selection of semantic descriptors is constraining the diversity search: it’s always the case no matter the representation space and it’s precisely a feature of the approach to give control to the experimenter on the kind of diversity it is aiming for. Nonetheless, we show that optimizing for diversity in this semantic space increases the diversity measured in other unrelated representation spaces (the three embedding spaces). For this reason, we argue that this approach is useful: it’s controllable, it’s easy to set-up, it allows to generate diversity at higher levels of abstraction and diversity that generalizes to other representation spaces.

---

### Author Response · Authors · 2023-11-17
**General Answer (part 2/2)**

## 2) About the evaluation metrics for diversity:

ACES assumes access to a semantic labeler whose features are defined by an experimenter. The one we use here is based on ChatGPT. As the confusion matrices show, it is imperfect, in the sense that it doesn’t align perfectly with human judgments. As such, and as noted by several reviewers, this is not a fundamental problem in the context of guiding the diversity search. It only poses a problem when it is used for evaluation. The problem lies in the fact that this semantic diversity metric measures diversity in a space that is somewhat mis-aligned with the one the experimenter cares about (the one resulting from an ideal puzzle labeler).

This is precisely for this reason that we propose other diversity metrics computed from three different embedding spaces that were not used in the training procedure. On its own, the fact that our method outperforms others on these unbiased evaluation metrics should be sufficient to make the argument that our method allows to generate more diversity than existing ones. Let us assume we were proposing a method optimizing for diversity in a given embedding space and showed that this method outperformed others in unrelated embedding spaces. Wouldn’t this be enough to claim that this method generates stronger diversity than existing ones? This is exactly the evaluation method that we report here. The semantic descriptor evaluation, in addition to the qualitative discussion of generated puzzles, brings additional elements to the study of the generated diversity, although we agree that, alone, it does not constitute a definitive argument.

**Reference:**
Etcheverry et al., 2020, Hierarchically Organized Latent Modules for Exploratory Search in Morphogenetic Systems

---

### Meta-Review · Area_Chair_L9Z4 · 2023-12-06

**Metareview:**

This paper proposes ACES, a prompting-based method to generate diverse programming puzzles to complement the existing Python programming puzzles dataset (P3). ACES prompts ChatGPT to map each puzzle in P3 to one of the ten manually defined semantic classes (semantic descriptors). New puzzles for each semantic class are generated by prompting ChatGPT using existing puzzles in that semantic class as few-shot exemplars. Experiments show that ACES generates a richer diversity of puzzles than existing methods. However, such diversity in the resulting puzzles cannot translate into better puzzle solving models when training those models on the sampled puzzle set.

**Strengths**

* The paper is clearly written with adequate technical details and evaluation results (XBPq)
* The proposed prompting-based approach is simple and effective, yielding a puzzle set with greater diversity (AUHG, ufAe).
* The base idea of stratified sampling based on a set of pre-defined example features sounds pretty general, and could potentially be applied to other machine learning problems that require a diverse set of synthetic samples (AUHG).
* The evaluation is extensive (ufAe), and the accompanying qualitative analysis shows that ACES is able to generate non-trivial puzzles (AUHG, ufAe)

**Weaknesses**

* The primary concern is about the amount of contribution made in this paper. While generating diverse programming puzzles sounds interesting, it’s unclear the actually *utility* of the generated puzzles (ufAe, AUHG), as authors show that ACES did not help in fine-tuning for the downstream task of puzzle solving.
* Another issue is that the semantic descriptors are manually curated (XBPq), without exploring automatic generation of those features, it’s questionable whether this approach could generalize to other applications without predefined sets of semantic classes (AUHG). On the other hand, relying on a fixed set of semantic descriptors could adversely affect sample diversity, and this might restrict the samples to those pre-defined semantic features (AUHG).
* It’s not clear if ChatGPT is well calibrated with human judgment when it’s used to automatically measure the diversity of the generated puzzles (AUHG, ufAe). Besides, there are no additional evaluation metrics for those puzzles other than sample diversity (ufAe)
* Missing discussion/comparison with existing synthetic data generation approaches that also consider sample diversity (ufAe)

**Justification For Why Not Higher Score:**

This paper introduces an approach to improve the diversity of synthetic samples for a very specific task of solving programming puzzles. It's not clear if the generated synthetic samples could be used to train better task-specific models. In addition, given the amount of domain-specific knowledge required to design semantic descriptors, it is not clear if the approach could be easily generalized to other applications. Therefore, the contribution of this paper is limited.

**Justification For Why Not Lower Score:**

N/A

---

### Decision · Program_Chairs · 2024-01-16

Reject